# Childhood gut microbiome is linked to internalizing symptoms at school age via the functional connectome

Francesca R. Querdasi [1,26] ✉, Jessica P. Uy [1,2,26], Jennifer S. Labus[3,4,5,6], Jia Xu [7], Neerja Karnani[7,8], Ai Peng Tan [7], Birit B. F. P. Broekman[7], Peter D. Gluckman [7,9], Yap Seng Chong[7,10,11], Helen Chen[12], Marielle V. Fortier[7], Lourdes Mary Daniel[13,14], Fabian Yap[15,16,17], Johan G. Eriksson[7,18,19,20], Shirong Cai[7], Mary Foong-Fong Chong[21], Jia Ying Toh[7], Keith M. Godfrey [22,23], Michael J. Meaney [24,25] & Bridget L. Callaghan[1]

The microbiome-gut-brain-axis plays a critical role in mental health. However, research linking the microbiome to brain function is limited, particularly during development, when tremendous plasticity occurs and many mental health issues, like depression and anxiety, initially manifest. Further complicating attempts to understand interactions between the brain and microbiome is the complex and multidimensional nature of both systems. In the current observational study (N = 55), we use sparse partial least squares to identify linear combinations of brain networks (brain signatures) derived from resting state fMRI scans at age 6 years that maximally covary with internalizing symptoms at age 7.5 years, before identifying microbe abundances (microbial profiles) derived from 16S rRNA sequencing of stool samples at age 2 years that maximally covary with those brain signatures. Finally, we test whether any early microbial profiles are indirectly associated with later internalizing symptoms via the brain signatures, highlighting potential microbial programming effects. We find that microbes in the Clostridiales order and Lachnospiraceae family are associated with internalizing symptoms in middle childhood through connectivity alterations within emotion-related brain networks.

The gut and brain are inextricably linked through a dynamic, bidirectional communication network known as the microbiome-gut-brain axis (MGBA). While this connection is increasingly recognized as pivotal to adult mental health[1], its role in shaping behavioral and psychological outcomes during development, when mental health problems first emerge[2–4], remains poorly understood. Early childhood is a rapid period of gastrointestinal microbiome maturation[5,6]. The microbiome reaches an adult-like configuration by around 3 years of age[5,7,8], after which point it remains relatively stable[9–11]. This early period may represent a sensitive window during which microbial signals shape key aspects of brain development[6,12]. Such early influences could have lasting effects by programming neurobiological systems in ways that confer vulnerability or resilience to later mental health outcomes, particularly during the high-risk period of middle childhood[2,3]. A developmental perspective on the MGBA is therefore essential for understanding how early microbial signals may contribute to lifelong trajectories of mental health.

Recent evidence that microbiome composition is associated with neurocognitive and behavioral outcomes during childhood, such as internalizing symptoms[13–15], underscores the importance of

understanding the MGBA's contribution to mental health during development. To date, however, most studies examining the MGBA in the context of typical development have focused on infancy and toddlerhood, with an emphasis on neurocognitive, rather than mental health, outcomes[16–19]. These studies have found that individual differences in microbial alpha diversity are associated with variation in the structure and function of brain regions and networks implicated in sensorimotor processing, language, and emotion[16–19]. For example, greater alpha diversity in infancy has been concurrently linked to increased fronto-parietal connectivity—a network implicated in cognitive control, which in turn was associated with increased infant negative emotionality[19]. Similarly, higher alpha diversity at age 1 year has been associated with lower functional connectivity between the left amygdala and thalamus, and between the anterior cingulate cortex and insula, as well as greater connectivity between the supplemental motor area and inferior parietal lobule[18]. Higher alpha diversity at age 1 year has also been linked with poorer cognitive performance (visual reception and expression language) and brain volume at later developmental stages—greater left precentral, right angular, and left amygdala volumes at age 2 years[16]—highlighting potential programming effects of early microbiota on later brain structure.

In studies examining specific microbial taxa, concurrent and prospective associations with neurodevelopment have also been reported. For instance, Carlson et al.[17] found that relative abundances of *Streptococcus* and *Staphylococcus* at age 1 month were concurrently negatively associated with amygdala volumes, and positively associated with medial prefrontal cortex (mPFC) volumes. In terms of potential programming effects, a higher abundance of *Bacteroides* at age 1 month was linked to smaller mPFC at age 1 year, whereas greater *Lachnospiraceae* abundance was prospectively associated with larger amygdala volume—an association that was reversed for *Enterobacteriaceae*[17]. Notably, in that study, there were no concurrent associations between specific bacterial taxa and brain volumes at age 1 year, suggesting cascading developmental effects, whereby earlier microbial composition influences later brain structure. In that study, concurrent and prospective associations between microbes and affective outcomes were also reported: higher abundances of *Dialister* and bacteria that are part of the Clostridiales order, at 1 year, were associated with reduced fear behavior at 1 year, and alpha diversity at 1 month was associated with increased fear behavior at 1 year. These findings align with the hypothesis, also supported by research in rodents[20], that early-life microbiota provide critical signaling inputs necessary for typical neurodevelopment, and that disruptions in these signals may confer risk for later psychopathology[6]. Despite these insights, it remains unclear whether—and how—early life microbial diversity and composition affect brain function later in development, during middle childhood, a high-risk period for mental health problems[4,21].

Of the developmental studies linking the microbiota to brain function, most have employed seed-based resting state functional connectivity (RSFC) analyses[16,18]. While use of RSFC is preferable in developing populations due to reduced task demands, the use of researcher-selected seeds in these analyses provides a limited view of brain function, and further, does not account for connectivity within and between canonical large-scale functional brain networks, which have been linked to neurocognitive behaviors and psychopathological outcomes in children[22–28]. For example, internalizing symptoms in children have been associated with alterations in RSFC within the default mode network (DMN), the ventral attention network (VAN), and between both DMN and striatal regions with cingulo-opercular (CON) networks[22]. Across development, a pattern of increasing within-network integration and between-network segregation supports the increasing specialization of network-related functions and efficient information processing—deviations from which are associated with psychopathology in youth[29–33]. As such, it is clear that investigating the impact of early microbiota on the development of brain network integration and segregation is an essential step for understanding the microbiota's contributions to psychopathology. However, such analyses are challenging due to the multivariate complexity of both microbiota and large-scale brain network data. Use of multivariate machine learning analysis techniques may help to solve this issue.

Multivariate machine learning approaches address key limitations of univariate neuroimaging analyses, including small effect sizes and the inability to identify multivariate combinations of brain patterns that relate to psychopathology[34]. Among these, canonical correlation analysis (CCA) and partial least squares (PLS) are widely used to reduce complex datasets into latent components that maximize shared variance—correlation (CCA) or covariance (PLS)[35,36]. PLS is particularly advantageous in situations when the sample size is similar to, or much smaller than, the number of variables in the dataset, and in situations where multicollinearity is present[35,37,38]. Incorporating sparsity into the PLS can facilitate interpretation and reduce overfitting by selecting a subset of relevant variables[35,37,38], making this approach (sparse PLS, or sparse partial least squares (sPLS)) especially robust to small sample sizes[39].

While multivariate machine learning techniques, including sPLS, are frequently used to compress microbiota data and identify its associations to other variables—including within a multi-omics framework[40], clinical outcomes[41], and with the brain[42,43]—these techniques have been underused in developing populations, which are often characterized by the small samples for which these approaches are particularly well-suited. Similarly, few studies investigating brain function in development have employed multivariate machine learning techniques, though a recent example using CCA was successful in identifying a connectome—comprising attentional, cognitive control (including DMN, frontoparietal, and salience), and subcortical networks—that was associated with cognition, behavioral/emotional outcomes, and diagnoses of mental disorders both concurrently and prospectively, i.e., 2 years after the neuroimaging data were collected[24]. In the current study, we advance the field by using sPLS to link early life microbiota to large-scale functional brain networks in middle childhood, and ultimately to internalizing symptoms later in childhood.

In the current study, we leveraged data from an ongoing longitudinal, observational cohort (the Growing Up in Singapore Towards Healthy Outcomes [GUSTO] study). Stool samples and resting state fMRI scans were collected from child participants at 2 and 6 years of age, respectively. At 7.5 years of age, caregivers completed questionnaires about children's behavioral problems. In previous analyses of data from this cohort, we demonstrated that maternal childhood maltreatment, maternal prenatal anxiety, and children's exposure to stressful life events were associated with distinct gastrointestinal microbiota profiles at 2 years of age, which were in turn associated with behavioral problems at 2 and 4 years of age[44]. In the current study, we investigate how gastrointestinal microbiota composition at 2 years of age relates to children's intrinsic functional brain network connectivity at 6 years of age, with these brain networks defined by their association with caregiver-reported internalizing problems at 7.5 years of age. We focus on internalizing problems for two reasons: First, internalizing symptoms during early-middle childhood increases risk for chronic and recurrent internalizing symptoms across the lifecourse[45,46], making this a particularly important symptom group to investigate for links to the microbiota. Second, recent evidence suggests that the associations between microbiota composition (alpha diversity and specific microbes) are more robustly associated with internalizing symptoms and somatic complaints than externalizing symptoms during the preschool years[14]. This multivariate symptom-centered approach to identify specific brain functional architecture relevant to mental health, before linking it to the microbiome, is a novel approach that can provide potential targets for future interventions. Furthermore, our examination of prospective links between early microbiota and brain networks/behavioral outcomes in middle childhood provides

insight into the potentially long reach of early microbial composition on child outcomes, especially during periods of risk for the emergence of internalizing symptoms[47,48].

We addressed our aims using a three-step approach. First, we used sPLS to identify combinations of resting brain networks (i.e., brain signatures) at age 6 years that maximally covaried with internalizing symptoms at age 7.5 years. We used the caregiver report of the child's internalizing symptoms as our primary indicator, as clinician assessment tends to agree more with parent-report than child-report in the age range of this study[49]. Based on previous research, we hypothesized that differences in RSFC within and between networks implicated in attention (e.g., ventral attention), cognitive and emotion regulation (e.g., DMN, frontoparietal, salience, cingulo-opercular), and subcortical areas (e.g., amygdala, hippocampus) would relate to caregiver-reported internalizing symptoms. Second, using these brain signatures as outcomes, we used sPLS again to identify combinations of specific microbial taxa (i.e., microbial profiles) at age 2 years that explained the most variability in those brain signatures. Further, we examined whether the compositional diversity of the gastrointestinal microbiota (alpha diversity) was associated with any brain signatures. We hypothesized that higher abundances of *Intestinibacter, Faecalibacterium, Coprobacillus*, and Lachnospiraceae, and lower abundance of *Veillonella* would be associated with brain signatures linked to lower caregiver-reported internalizing symptoms, as they have been previously associated with caregiver-reported mental health at age 2 or 4 years in this cohort in the hypothesized directions[44]. Finally, to statistically evaluate a conceptual model in which the connection between the microbiota and mental health is mediated by the brain, we tested whether brain signatures at age 6 years mediated associations between microbiota features (i.e., microbial profiles and alpha diversity) at age 2 years and caregiver-reported internalizing symptoms at age 7.5 years. For all analyses, we covaried for child sex assigned at birth, gestational age, birthweight, mean framewise displacement (motion in fMRI), delivery mode, maternal education, and diet.

## Results

### Descriptive statistics

The analytic sample for this study included $N = 55$ children who contributed a usable stool sample at age 2 years, good-quality fMRI data at age 6 years, and whose caregiver completed an assessment of their caregiver-reported internalizing problems at age 7.5 years. Of those, $N = 43$ who provided data on all covariates were included in analyses that controlled for covariates. Demographics, descriptive statistics for key study variables, and descriptive statistics for covariates, for the analytic sample and the sample of children excluded due to poor quality fMRI data ($N = 55$) are shown in Tables 1, 2, and 3, respectively. The only difference between the included and excluded samples was that excluded participants had higher mean framewise displacement on average, which was the reason for their exclusion (Table 3).

### Two brain signatures maximally covaried with caregiver-reported internalizing symptoms

Two participants were identified as multivariate outliers that could be driving component results and were univariate outliers on connectivity within the striatal-orbitofrontal-amygdala (SOFA) network and connectivity between the SOFA and medial temporal lobe (MTL) networks. Thus, those two participants' connectivity values for SOFA and SOFA-MTL were winsorized, and model tuning was repeated. The final best-fitting model consisted of two orthogonal brain network components (Component 1: 20 networks, 12 of which had high stability; Component 2: 15 networks, 9 of which had high stability). See Supplementary Table 2 for all variable loading, Variable Importance in the Projection (VIP), and stability values.

To validate these components, we statistically tested their association with caregiver-reported internalizing symptoms with and

without controlling for covariates. Component 1 was correlated with caregiver-reported internalizing symptoms ($r = 0.47$, $p = 0.00012$) and was significantly positively related to internalizing symptoms when controlling for covariates ($\beta = 0.52$, SE = 0.23, $p = 0.036$; Fig. 1A). The top 4 highest magnitude loadings on Component 1 all had high stability and included positive and negative loadings. Positive loadings (i.e., stronger connectivity within and between these networks was positively associated with caregiver-reported internalizing symptoms) were within-network connectivity in the SOFA network and between-network connectivity between the SOFA and MTL networks. Negative loadings (i.e., weaker connectivity within and between these networks were associated with more caregiver-reported internalizing symptoms) were within-network connectivity in the SAL network and between-network connectivity between the SAL and parietomedial network (PMN). We therefore called this component the SOFA, MTL, SAL, PMN Network Connectivity Brain Signature. See Fig. 2 for a visualization of the networks in this signature.

Component 2 was correlated with caregiver-reported internalizing symptoms ($r = 0.38$, $p = 0.012$) and was significantly positively related to caregiver-reported internalizing symptoms when controlling for covariates ($\beta = 0.52$, SE = 0.21, $p = 0.019$; Fig. 1C). The top 4 highest magnitude loadings on Component 2, all of which had high stability, were all positive loadings (i.e., stronger connectivity within and between these networks was positively associated with caregiver-reported internalizing symptoms). These loadings included between-network connectivity between the SOFA and visual networks, between the SOFA and DMN, between the VAN and dorsal somatomotor network (SMD), and between the SOFA and FPN networks. We therefore called this component the SOFA Between Network Connectivity Brain Signature. See Fig. 3 for a visualization of networks in this signature.

### Three microbial profiles maximally covaried with brain signatures

**Microbial abundances associated with SOFA, MTL, SAL, PMN Network Connectivity Brain Signature.** There was one multivariate outlier, which was also a univariate outlier on *Coprococcus*, *Anaerobutyricum*, and *Weissella* abundance; abundances of those genera were winsorized, and tuning was repeated. The final best-fitting model consisted of 1 microbiota component with 5 genera (three of which had high stability). See Fig. 1B–E for visualization of loadings for variables with VIP > 1 and stability ≥ 0.8, and Supplementary Table 3 for loadings, VIP, and stability values for all variables on each component.

To validate the component, we statistically tested its association with the SOFA, MTL, SAL, PMN Network Connectivity Brain Signature with and without controlling for covariates. The component was correlated with the SOFA, MTL, SAL, PMN Network Connectivity Brain Signature ($r = 0.51$, $p = 0.0000071$) and was also significantly positively related to this brain signature when controlling for covariates ($\beta = 0.46$, SE = 0.21, $p = 0.031$). Of the five genera that loaded onto this component, three genera had high stability and included positive loadings for *Eubacteriaceae* abundance (i.e., higher abundance of these genera was associated with higher scores on the SOFA, MTL, SAL, PMN Network Connectivity Brain Signature), and negative loadings for *Anaerobutyricum* and *Dialister* abundance (i.e., higher abundance of these genera was associated with lower scores on the SOFA, MTL, SAL, PMN Network Connectivity Brain Signature). We called this component Microbial Profile 1.

**Microbial abundances associated with SOFA Between Network Connectivity Brain Signature.** There were no multivariate outliers. The final best-fitting model consisted of 2 microbiota components (Component 1: 5 genera, 2 of which had high stability; Component 2: 20 genera, 13 of which had high stability).

To validate the components, we statistically tested their associations with the SOFA Between Network Connectivity Brain Signature

**Table 1 | Demographics of the study sample**

| Measure[a] | Analytic sample (N = 55) | Participants excluded in fMRI QC (N = 55) | Significant difference between the analytic and excluded samples |
|---|---|---|---|
| Ethnicity | | | No |
| Chinese | 40 (73%) | 41 (75%) | |
| Malay | 10 (18%) | 9 (16%) | |
| Indian | 5 (9%) | 5 (9%) | |
| Child sex | | | No |
| Male | 23 (42%) | 28 (51%) | |
| Female | 32 (58%) | 27 (49%) | |
| Age stopped breastfeeding | | | No |
| <1 month | 0 (0%) | 13 (24%) | |
| 1 month to 3 months | 8 (14%) | 7 (13%) | |
| 3 months to 6 months | 10 (18%) | 10 (18%) | |
| 6 months to 12 months | 13 (24%) | 12 (22%) | |
| >12 months | 12 (22%) | 13 (24%) | |
| Missing | 12 (22%) | 0 (0%) | |
| Monthly income per household member (SGD) | 1345.85 (846.05); 124.88–3000.00+ | 1387.87 (834.90); 83.25–3000.00+ | No |
| Below poverty cutoff[b] | 13 (24%) | 11 (20%) | |
| Missing | 4 (7%) | 3 (5%) | |
| Maternal mental health (at child age 6 years) | | | |
| Maternal depressive symptoms (raw scores) | 7.61 (8.59); 0–36 | 9.83 (8.67); 0–38 | No |
| Minimal | 37 (67%) | 36 (65%) | |
| Mild | 4 (7%) | 6 (11%) | |
| Moderate | 4 (7%) | 2 (4%) | |
| Severe | 1 (2%) | 3 (5%) | |
| Missing | 9 (16%) | 8 (15%) | |
| Maternal anxiety symptoms (raw scores) | 33.74 (8.66); 20–52 | 34.88 (10.31); 21–57 | No |
| Below the clinical cutoff | 37 (67%) | 34 (62%) | |
| Above clinical cutoff | 10 (18%) | 14 (25%) | |
| Missing | 8 (15%) | 7 (13%) | |
| Other medication use in the past 6 months (at the child's age of 2 years) | | | No |
| None | 8 (15%) | 11 (20%) | |
| 1 | 5 (9%) | 8 (15%) | |
| 2 | 5 (9%) | 8 (15%) | |
| 3 or more | 37 (67%) | 28 (51%) | |
| Duration of medication use (days) | 4.82 (4.00); 1–45 | 5.26 (4.90); 1–30 | |

[a]Mean (SD); Range for continuous variables and N (%) for categorical variables.
[b]While Singapore does not have an official income threshold defining the poverty line, a family income of $1999, or a monthly income per member of $650, is commonly used in government agencies to indicate poverty[104].

with and without controlling for covariates. Component 1 was correlated with the SOFA Between Network Connectivity Brain Signature ($r = 0.58$, $p = 0.0000027$) and was also significantly positively related to this brain signature when controlling for covariates ($\beta = 0.48$, SE = 0.21, $p = 0.032$). Of the five genera that loaded onto this component, two had high stability and included positive loadings for *Eubacterium* and *Terrisporobacter* abundance (i.e., higher abundance of these genera was associated with higher scores on the SOFA Between Network Connectivity Brain Signature). We named this component Microbial Profile 2.

Component 2 was correlated with the SOFA Between Network Connectivity Brain Signature ($r = 0.67$, $p = 0.000017$) and was also significantly positively related to the brain signature when controlling for covariates ($\beta = 0.47$, SE = 0.19, $p = 0.020$). Of the 20 genera that loaded onto this component, 13 had high stability. The top 4 highest magnitude, stable loadings for this component were positive loadings

for *Veillonella* and *Intestinibacter* abundance (i.e., higher abundance of these genera was associated with higher scores on the SOFA Between Network Connectivity Brain Signature), and negative loadings for *Catenibacterium* and *Fusicatenibacter* abundance (i.e., higher abundance of these genera was associated with lower scores on the SOFA Between Network Connectivity Brain Signature). We named this component Microbial Profile 3. See Supplementary Table 3 for names, loadings, and VIP values for the other stable genera.

**Faith's phylogenetic diversity was positively associated with the SOFA Between Network Connectivity Brain Signature, but was not associated with the SOFA, MTL, SAL, PMN Network Connectivity Brain Signature**

Alpha diversity in the gut microbiota was not associated with scores on the SOFA, MTL, SAL, PMN Network Brain Signature, controlling for covariates: Shannon diversity ($\beta = 0.14$, SE = 0.27, $p = 0.60$), Pielou

**Table 2 | Descriptive statistics for key study variables**

| Measure[a] | Analytic sample (*N* = 55) | Participants excluded in fMRI QC (*N* = 55) | Significant difference between the analytic and excluded samples |
|---|---|---|---|
| Internalizing symptoms (raw scores) | 5.44 (4.65); 0–19 | 6.09 (6.24); 0–29 | No |
| Clinical symptom levels (T-score > 63) | 6 (11%) | 7 (13%) | |
| Borderline symptom levels (T-score: 60–63) | 3 (5%) | 5 (9%) | |
| Not clinically significant symptom levels (T-score <60) | 46 (84%) | 43 (78%) | |
| Anxiety symptoms (raw scores) | 2.25 (1.85); 0–7 | 2.47 (2.32); 0–9 | No |
| Depressive symptoms (raw scores) | 1.42 (1.58); 0–9 | 1.71 (2.57); 0–14 | No |
| Externalizing symptoms (raw scores) | 5.69 (5.28); 0–24 | 6.35 (6.08); 0–30 | No |
| Clinical symptom levels (T-score >63) | 3 (5%) | 4 (7%) | |
| Borderline symptom levels (T-score: 60–63) | 3 (5%) | 6 (11%) | |
| Not clinically significant symptom levels (T-score <60) | 49 (90%) | 45 (82%) | |
| Alpha diversity | | | |
| Shannon Index | 3.26 (0.77); 1.57–5.14 | 3.25 (0.67); 0.80–4.48 | No |
| Observed Features | 81.44 (24.50); 48–173 | 76.04 (19.43); 38–141 | No |
| Pielou Evenness | 0.51 (0.10); 0.28–0.69 | 0.52 (0.09); 0.14–0.64 | No |
| Faith's phylogenetic diversity | 5.73 (1.76); 3.24–12.65 | 5.45 (1.82); 2.94–13.39 | No |

[a]Mean (SD); Range for continuous variables and *N* (%) for categorical variables.

**Table 3 | Descriptive statistics for covariates**

| Measure[a] | Analytic sample (*N* = 55) | Participants excluded in fMRI QC (*N* = 55) | Significant difference between the analytic and excluded samples |
|---|---|---|---|
| Birth method | | | No |
| Vaginal | 38 (69%) | 40 (73%) | |
| Cesarean section | 17 (31%) | 15 (27%) | |
| Gestational age at birth (weeks) | 39.00 (1.13); 36.71–41.14 | 39.05 (1.20); 35.71–41.00 | No |
| Birthweight (kg) | 3.16 (0.40); 2.45–4.05 | 3.18 (0.44); 1.69–4.14 | No |
| Macronutrients in the diet | | | |
| Protein (% energy) | 15.74 (3.72); 9.07–26.52 | 16.02 (2.33); 11.04–20.97 | No |
| Total fat (% energy) | 29.53 (6.52); 14.35–41.42 | 28.63 (5.15); 20.39–39.84 | No |
| Carbohydrates (% energy) | 54.73 (7.00); 44.32–75.66 | 55.35 (5.335); 41.36–65.01 | No |
| Monounsaturated fat (MUFA) (% energy) | 34.09 (10.78); 9.68–54.03 | 34.04 (9.84); 8.22–55.89 | No |
| Polyunsaturated fat (PUFA) (% energy) | 33.36 (15.88); 12.85–72.12 | 34.40 (14.20); 16.41–75.20 | No |
| Saturated fat (% energy) | 32.54 (10.09); 12.96–50.94 | 31.57 (9.70); 9.80–50.11 | No |
| Fiber (g per 1000 kcal) | 5.97 (3.39); 1.41–19.03 | 5.74 (2.40); 0.49–10.55 | No |
| Mean framewise displacement (FD) | 0.09 (0.04); 0.04–0.19 | 0.43 (0.30); 0.06–1.30 | Yes |
| Maternal highest level of education achieved | | | No |
| University degree (BA) or higher | 16 (29%) | 19 (35%) | |
| GCE A Levels/Polytechnic/Diploma | 19 (35%) | 11 (20%) | |
| ITE/NITEC | 0 (0%) | 6 (11%) | |
| Secondary (GCE O/N Levels) | 12 (22%) | 10 (18%) | |
| Primary (PSLE) | 3 (5%) | 3 (5%) | |
| Missing | 5 (9%) | 6 (11%) | |
| Antibiotic use (at child age of 2 years) | | | No |
| Yes | 7 (13%) | 5 (9%) | |
| No | 48 (87%) | 50 (91%) | |
| Probiotic use (at child age of 2 years) | | | No |
| Yes | 3 (5%) | 2 (4%) | |
| No | 52 (95%) | 53 (96%) | |

[a]Mean (SD); Range for continuous variables and *N* (%) for categorical variables.

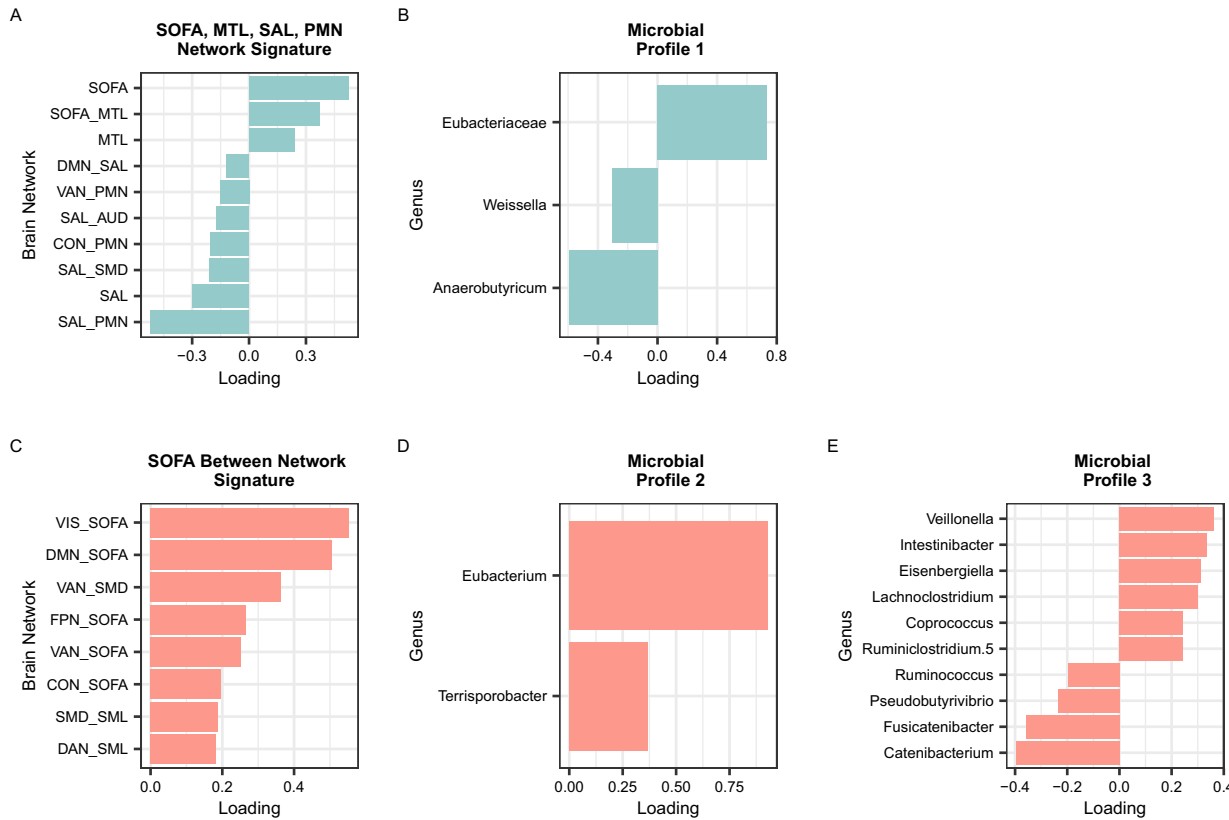

**Fig. 1 | Component loadings for brain signatures and microbial profiles. A** SOFA, MTL, SAL, PMN Network Signature. **B** Microbial Profile 1. **C** SOFA Between Network Signature. **D** Microbial Profile 2. **E** Microbial Profile 3. Colors (blue or salmon) and rows correspond to the brain signature that the microbial profile was derived from (blue = SOFA, MTL, SAL, PMN Network Connectivity Brain Signature; salmon = SOFA Between Network Connectivity Brain Signature). Only variables with VIP > 1 are shown. SOFA = striatal-orbitofrontal-amgydalar, MTL = medial temporal lobe, DMN = default mode network, SAL = salience, PMN = parietomedial network, VAN = ventral attention network, AUD = auditory, CON = cingulo-opercular, SMD = somatomotor dorsal. For microbes that were unable to be classified to the genus level, the family name is shown. Source data are provided as a Source data file.

evenness (β = 0.16, SE = 0.27, p = 0.55), observed features (β = 0.02, SE = 0.23, p = 0.94), or Faith's phylogenetic diversity (β = 0.11, SE = 0.22, p = 0.64).

Higher Faith's phylogenetic diversity in the gut microbiota was associated with higher scores on the SOFA Between Network Connectivity Brain Signature, controlling for covariates (β = 0.37, SE = 0.12, p = 0.008, q = 0.024; Fig. 4). The other alpha diversity metrics were not associated with scores on the SOFA Between Network Connectivity Brain Signature: Shannon diversity (β = 0.23, SE = 0.18, p = 0.22), Pielou evenness (β = 0.20, SE = 0.18, p = 0.29), observed features (β = 0.17, SE = 0.22, p = 0.45).

**Gut Microbial Profile 3 was positively associated with caregiver-reported internalizing symptoms through the SOFA Between Network Connectivity Brain Signature**

Microbial Profile 3 was indirectly positively associated with caregiver-reported internalizing symptoms at age 7.5 years through higher scores on the SOFA Between Network Connectivity Brain Signature, controlling for covariates (β = 0.32, SE = 0.17, 95% CI: [0.0075, 0.68]). There was no direct (β = −0.24, SE = 0.20, 95% CI: [−0.65, 0.16]) or total (β = 0.01, SE = 0.25, p = 0.96) effect of Microbial Profile 3 on internalizing symptoms, controlling for covariates (Fig. 5).

Microbial Profile 1 was not associated with caregiver-reported internalizing symptoms at age 7.5 years, directly (β = 0.18, SE = 0.20,

95% CI: [−0.22, 0.59]), indirectly through the SOFA, MTL, SAL, PMN Network Connectivity Brain Signature (β = 0.16, SE = 0.13, 95% CI: [−0.061, 0.43]), or overall (i.e., the total effect; β = 0.34, SE = 0.20, p = 0.10), controlling for covariates (Supplementary Table 5). Similarly, Microbial Profile 2 was not associated with caregiver-reported internalizing symptoms at age 7.5 years directly (β = 0.29, SE = 0.24, 95% CI: [−0.20, 0.79]), indirectly through the SOFA Between Network Connectivity Brain Signature (β = 0.17, SE = 0.19, 95% CI: [−0.11, 0.65]), or overall (i.e., the total effect; β = 0.46, SE = 0.24, p = 0.065), controlling for covariates (Supplementary Table 5).

There was no total effect of Faith's phylogenetic diversity on caregiver-reported internalizing symptoms, controlling for selected covariates (β = 0.21, SE = 0.20, p = 0.31). Faith's phylogenetic diversity was also not related to internalizing symptoms indirectly through the SOFA Between Network Connectivity Brain Signature (β = 0.19, SE = 0.14, 95% CI: [−0.084, 0.48], and there was also no direct effect of Faith's phylogenetic diversity on caregiver-reported internalizing symptoms in this model (β = 0.013, SE = 0.13, 95% CI: [−0.26, 0.28]; Supplementary Table 4). Although Faith's phylogenetic diversity was not associated with the SOFA, MTL, SAL PMN Network Connectivity Brain Signature, we still tested a mediation for Faith's phylogenetic diversity on caregiver-reported internalizing symptoms via this signature for completeness, and results are presented in Supplementary Table 5.

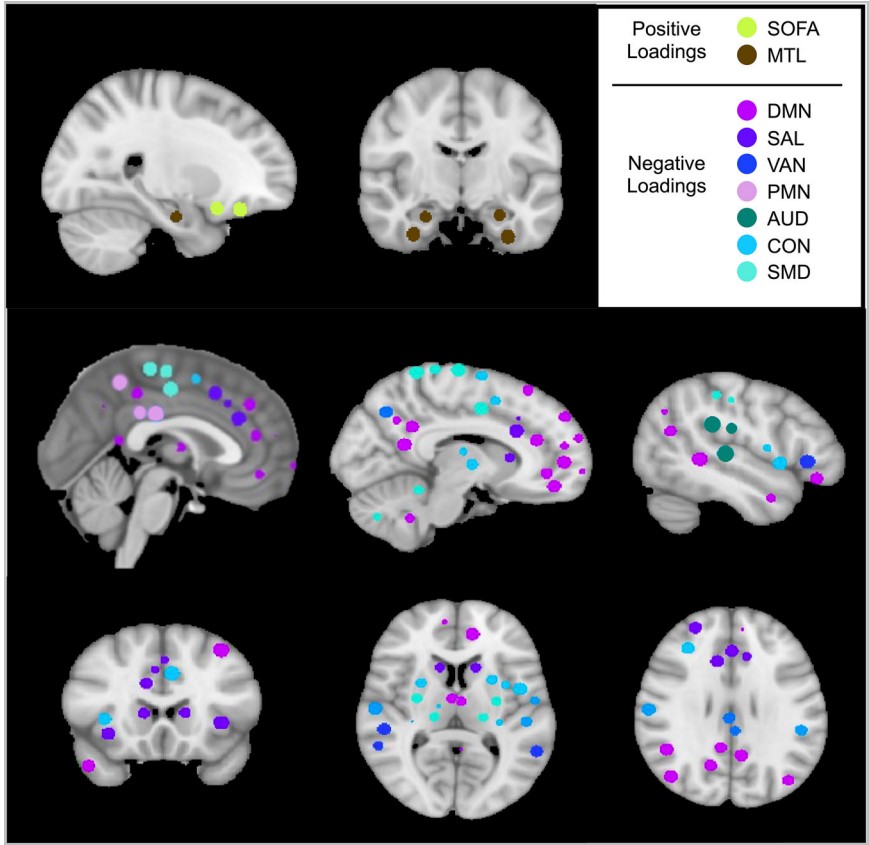

**Fig. 2 | Regions of interest in the networks with loadings onto the SOFA, MTL, SAL, PMN Network Connectivity Brain Signature.** Regions of interest and network assignments were derived from the Seitzman et al.[90] atlas. SOFA striatal orbitofrontal amygdalar, MTL medial temporal lobe, DMN default mode, SAL salience, VAN ventral attention, PMN parietomedial, AUD auditory, CON cingulo-opercular, SMD dorsal somatomotor.

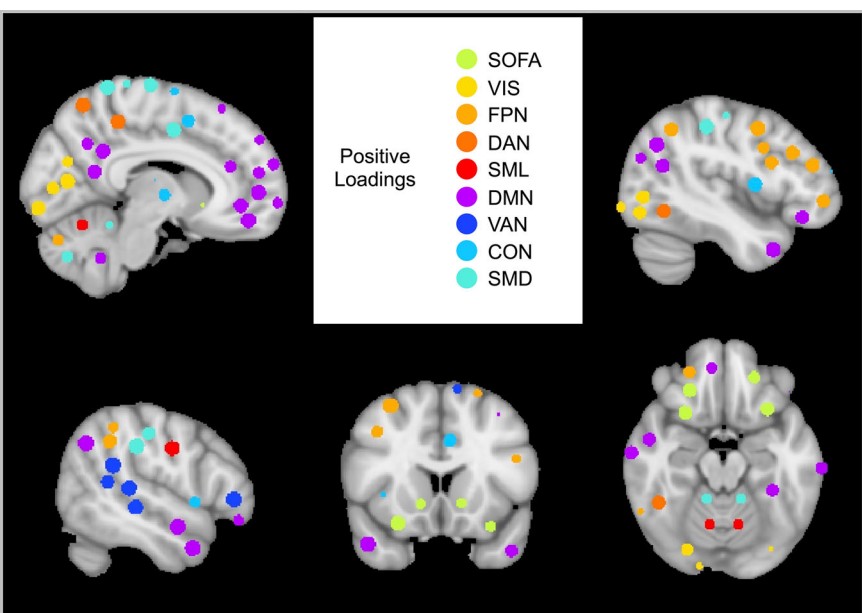

**Fig. 3 | Regions of interest in the networks with loadings onto the SOFA Between Network Connectivity Brain Signature.** Regions of interest and network assignments were derived from the Seitzman et al.[90] atlas. SOFA = striatal orbitofrontal amygdalar, VIS = visual, FPN = frontal parietal, DAN = dorsal attention, SML = lateral somatomotor, DMN = default mode, VAN = ventral attention, CON = cingulo-opercular, SMD = dorsal somatomotor.

## Specificity of brain signatures and microbiota profiles to caregiver-reported internalizing symptoms

To test specificity of links between Brain Signatures and Microbial Profiles to caregiver-reported internalizing symptoms broadly, we examined all identified Brain Signatures and Microbial Profiles and their links with caregiver reported anxiety and depressive symptoms separately at 7.5 years of age (which go into the internalizing symptom score), as well as with caregiver-reported externalizing symptoms at 7.5 years of age, and with child self-reported depression symptoms and anxiety symptoms at 8.5 years of age. We found that the SOFA, MTL,

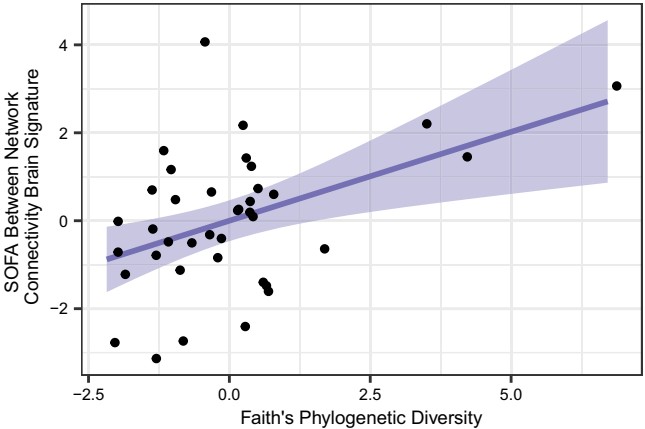

**Fig. 4 | Significant association between Faith's phylogenetic diversity and SOFA Between Network Connectivity Brain Signature scores.** Each dot represents a participant. Faith's Phylogenetic Diversity and SOFA Between Network Connectivity Brain Signature scores are shown after partialling out variance accounted for by covariates. Predictors in the model included Faith's phylogenetic diversity, child sex, gestational age at birth, birthweight, delivery mode, mean FD, maternal education and consumption of fiber, total fat and polyunsaturated fat. The dark purple line represents the predicted best-fit function predicting *x* from *y* using a linear model, and the lighter purple shaded area around the line is the 95% CI level for that predicted best-fit function. Dots have been jittered horizontally (along the *x*-axis) by 0.3 to increase visibility of individual data points. Source data are provided as a Source data file.

SAL, PMN Network Connectivity Brain Signature was positively associated with caregiver-reported depression, anxiety, and externalizing symptoms at 7.5 years of age, as well as with child self-reported depressive symptoms at 8.5 years of age. In other words, this Brain Signature was not specific to the caregiver-reported internalizing symptoms upon which it was defined. The SOFA Between Network Connectivity Brain Signature was associated with child self-reported depressive symptoms at 8.5 years of age, suggesting it too was not completely specific to the caregiver-reported internalizing symptoms upon which it was defined. Similarly, Microbial Profile 1 was positively associated with caregiver-reported externalizing symptoms at 7.5 years of age, again suggesting some degree of generalizability beyond caregiver-reported internalizing symptoms. In contrast, Microbial Profiles 2 and 3 were not associated with any additional symptom measures, which suggests that the indirect association between Microbial Profile 3 and caregiver-reported internalizing symptoms reported above (mediated by the SOFA Between Network Connectivity Brain Signature) is indeed specific to that outcome measure. Methods and full results are presented in the SI.

## Predicted microbiota functional profiles that maximally covaried with brain signatures

To better understand relations between the predicted functional potential of the microbiota and identified brain signatures, we estimated the abundance of functional pathways from the 16S sequencing data using PiCRUST 2, and then performed sPLS analyses with that predicted functional data as input. We found that one predicted functional profile covaried most strongly with the SOFA, MTL, SAL, PMN Network Connectivity Brain Signature, and remained significantly associated with its corresponding brain signature when controlling for covariates. The pathway that had high stability from this component, which also had the strongest loading, was PWY-6270, isoprene biosynthesis I (negative loading). We also found two predicted functional profiles covaried most strongly with the SOFA Between Network Connectivity Brain Signature, one of which remained significantly associated with its corresponding brain signature when controlling for covariates. Pathways from these profiles with the highest loadings, and high stability, included the pentose phosphate pathway and superpathway of glucose and xylose degradation (negative loadings), and

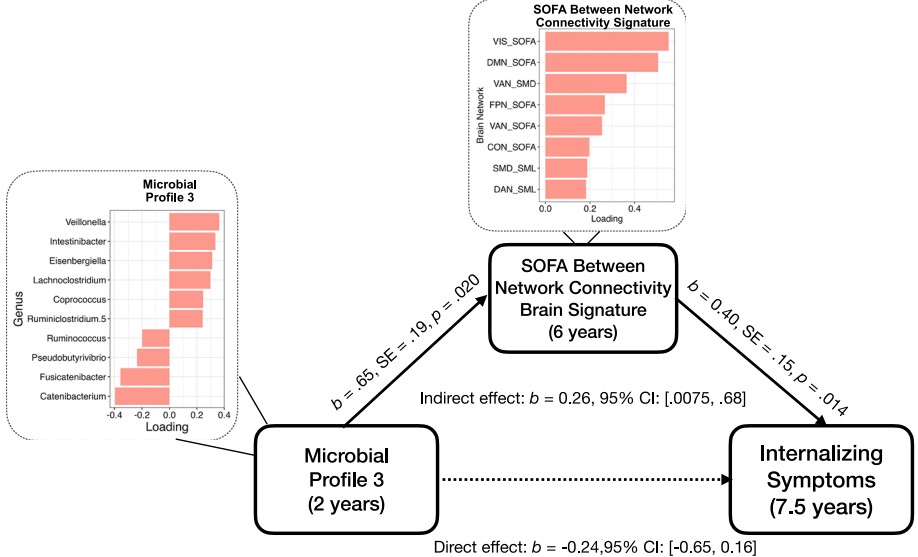

**Fig. 5 | Diagram summarizing results from the mediation analysis testing the effect of Microbial Profile 3 on child internalizing symptoms through the SOFA Between Network Connectivity Brain Signature.** Covariates include child sex, gestational age at birth, birthweight, delivery mode, mean FD, maternal education, and consumption of fiber, total fat and polyunsaturated fat for both the *a* and *b* paths.

cob(II)yrinate a,c-diamide biosynthesis I pathway, NAD salvage pathways V and I, and UPD-N-acetyl-D-glucosamine biosynthesis I pathway (positive loadings). There were no significant total, direct, or indirect associations (via the corresponding brain signature) between the functional profiles and caregiver-reported internalizing symptoms at age 7.5 years. Full methods and results are presented in the Supplementary Methods and Notes, respectively.

## Discussion

A small but growing body of research has investigated the role of the MGBA in developmental psychopathology. This study builds on past work by using a top-down approach to first identify internalizing symptom-relevant network brain signatures, and then identify gut microbiota taxa abundance profiles that explain variance in those brain signatures. These brain signatures and microbial profiles then informed mediation models where we tested indirect links between gut microbiota and internalizing symptoms via brain networks. We found that variance in internalizing symptoms at age 7.5 years was best explained by two orthogonal brain network connectivity signatures at age 6 years. These signatures were best explained by three gut microbiota abundance profiles at age 2 years. In turn, one of the microbial profiles was indirectly related to internalizing symptoms via its corresponding brain signature, suggesting a longitudinal pathway through which the gut microbiota in early life may be associated with later emerging functional connectomes implicated in childhood internalizing symptoms.

The two orthogonal brain signatures identified as maximally covarying with internalizing symptoms included several networks that have previously been associated with mental health and cognition. The first of the identified signatures, SOFA, MTL, SAL, PMN Network Connectivity Brain Signature, was characterized by higher connectivity within each of the SOFA and MTL networks, higher connectivity between the SOFA and MTL networks, weaker connectivity within the SAL network, and weaker connectivity between the SAL and PMN networks. Consistent with those results, alterations in RSFC between the amygdala and prefrontal regions (both parts of the SOFA network) have been found in youth depression[50] and anxiety[51]. Though some studies have reported the opposite association (as reviewed in Rakesh et al.[52]). Connectivity within the SAL, and between the amygdala and MTL, has also been associated with psychopathology in adolescence, though the direction of associations has varied across studies and disorders[53-57]. Consistent with this study, weaker connectivity between the SAL and precuneus (a region within the PMN) has been associated with higher trait anxiety in adolescents[58]. Considered in the context of prior research, this brain signature suggests disrupted connectivity within the SOFA, MTL, SAL, and PMN networks in middle childhood may be implicated in internalizing symptoms slightly later in development.

The second of the identified internalizing-associated brain signatures—the SOFA Between Network Connectivity Brain Signature— was best characterized by higher connectivity between the SOFA and a variety of other networks. This signature is consistent with past research examining neural correlates of internalizing symptoms. Specifically, amygdala (part of the SOFA network) hyperconnectivity with neurocognitive networks, including the CON, VAN, and DMN, has frequently been associated with psychopathology at different developmental stages[59], including associations with transdiagnostic mental health symptoms[23], as well as specifically with major depressive disorder[60], and irritability[61]. Moreover, SOFA hyperconnectivity is thought to reflect disruption in neurocircuitry underlying cognitive control and emotion regulation[23]. Overall, the connectivity profile observed within this signature suggests that increased subcortical connectivity with a range of cognitive and perception networks is associated with internalizing symptoms in middle childhood.

Critical to the interpretation of internalizing symptom-associated brain signatures, in the current study, only a minority of children exhibited symptoms at clinical or borderline levels, consistent with prevalence rates in the general population[62,63]. However, because symptom severity exists along a continuum, focusing exclusively on clinically significant distress (i.e., scores above diagnostic cutoffs) may overlook meaningful variation in functioning. Subclinical symptoms, particularly of depression, have been shown to increase the risk of later diagnoses of major depressive disorder—potentially through pathways involving cognitive dysfunction and anhedonia[64-66]. Given the younger age of this sample, which precedes the median age of onset for disorders such as anxiety (11 years) and depression (30 years), examining subthreshold symptoms may offer valuable insights into early indicators of risk and future outcomes[67].

With respect to microbial profiles, which explained maximal variance in the brain signatures, we saw that they too had been associated with internalizing symptoms in the extant literature. We identified three orthogonal microbial profiles, the first of which was linked to the SOFA, MTL, PMN Network Connectivity signature. This profile (Microbial Profile 1) was characterized by positive stable loading for the Eubacteriaceae family, and negative stable loadings for Weissella and Anaerobutyricum. Interestingly, both Weissella and Anaerobutyricum have been linked to better mental and physical health outcomes via the anti-inflammatory effects of their metabolites[68-70]. Taken together, these findings suggest that connectivity within and between the SOFA, MTL, SAL, and PMN networks might be best explained by microbes that modulate inflammation.

The second and third microbial profiles were both linked to the SOFA Between Network Connectivity Brain Signature. Microbial profile 2 was characterized by positive loadings for Eubacterium and Terrisporobacter genera. Interestingly, both of these genera have been linked to negative health outcomes in other populations. For instance, higher Eubacterium abundance has been related to poorer physical health in clinical child samples[71,72], and Terrisporobacter was found to be higher in adults with irritable bowel syndrome than in control adults without those symptoms[73]. However, Eubacterium species also metabolize cholesterol, contributing to lower cholesterol levels in the blood[74], which is a positive health outcome. Nonetheless, on the whole, research suggests that SOFA Between Network Connectivity might be best explained, in part, by microbes linked to physical health symptoms, which are often comorbid with internalizing symptoms[75].

Microbial profile 3, also linked to the SOFA Between Network Connectivity Brain Signature, was largely characterized by genera from the Lachnospiraceae family (6 out of the 13 stable genera in this profile are members of the Lachnospiraceae family; see Supplementary Table 3 for full taxonomic classification). Interestingly, in a small proof-of-concept study of school-age children, another microbe from the Lachnospiraceae family was positively associated with PFC reactivity to fearful faces[76]. Of the top 4 highest loading and stable microbes in this component, higher abundance of Veillonella (positive loading on Microbial Profile 3) was previously related to more sleep problems in early childhood in this cohort[44], and in other cohorts it has been associated with more internalizing symptoms at school age[14], emotion regulation difficulties in infancy[77], and stronger fear behavior in infancy[17]. Veillonella abundance has also been linked to more severe anxiety symptoms in adults[78]. Consistent with their negative loading onto a brain component associated with internalizing symptoms in this study, Fusicatenibacter and Catenibacterium have been linked to less severe depressive symptoms in adults[79,80]. Only one study, also in this cohort, has reported an association between Intestinibacter abundance and child mental health (caregiver-reported internalizing symptoms at age 4 years and total behavior problems at age 2 years), but in the opposite direction as its association with brain networks in this study[44]. The differing directions of association across studies and

the lack of prior research on this genus suggest that future research should continue to investigate the relation between *Intestinibacter* and mental health in childhood. Overall, the SOFA Between Network Connectivity Brain Signature also appears to be characterized, in part, via internalizing symptom-associated microbes.

Interestingly, the microbes contained within the three identified microbiota profiles shared similarities at higher taxonomic levels. Twelve out of the fifteen microbes across the three profiles with VIP > 1 were in the Clostridiales order, with five of those being in the Lachnospiraceae family within the Clostridiales order (see Table S3). Though the direction of association with internalizing symptoms differed across these microbes, the strong representation of the Clostridiales order and Lachnospiraceae family parallels findings from other work on young adults with depression[81]. Interestingly, changes within these microbe populations have also been associated with inflammatory response to acute laboratory stressors in adults[82], and are often seen in developmental work on early life adversity and the gut microbiota[44,76,83], suggesting that they may be particularly susceptible to modulation by stress and may contribute to the link between exposure to adversity and development of internalizing symptoms.

While our taxonomic analyses suggested that microbes involved with inflammation, physical, and mental health symptoms were explaining most variance in the internalizing symptom-associated brain signatures, supplemental analyses that identified profiles of predicted microbiome functional pathways most strongly linked to the derived brain signatures found the strongest associations with pathways involved in cellular energy metabolism (nicotinamide adenine dinucleotide salvage and phosphate pathways)[84,85]. Future research using functional potential derived from shotgun metagenomics is needed to verify these findings, given the established limitations of inferring functional potential from amplicon sequencing data[86].

Beyond looking at microbial profiles linked with brain signatures, we also evaluated associations between gut microbiota alpha diversity and brain signatures, finding that Faith's phylogenetic diversity at 2 years was positively associated with the SOFA Between Network Connectivity Brain Signature at 6 years. While this association appeared to be driven by three participants who were high on both the brain signature and Faith's phylogenetic diversity, those individuals were part of a larger group of children who had higher Faith's diversity values in this cohort, suggesting that they represent a real subset of individuals in the sample.

One very interesting feature of this study was the fact that we saw gut microbiota profiles that were linked not only to internalizing symptom-relevant brain networks, but also to internalizing symptoms themselves, through a brain network-mediated pathway. Specifically, we found that the SOFA Between Network Connectivity Brain Signature mediated the relationship between Microbial Profile 3 and internalizing symptoms. Interestingly, features comprising Microbial Profile 3, and the SOFA Between Network Connectivity Brain Signature, have been associated with internalizing symptoms in other studies[17,23,44,59–61,76,77,79,80], suggesting they may constitute an important brain-gut pathway involved in internalizing symptom expression. Our finding highlights the potential utility of the machine learning approach used in this study to accelerate the discovery of possible microbial biomarkers of later emerging mental health problems. Future studies can focus attention on these microbial features in assessments of links to internalizing problems and disorders, and ultimately in studies testing microbial interventions for internalizing symptoms.

A notable strength of this study is that it included an assessment of internalizing symptoms along with measures of the gut microbiota and brain, allowing us to anchor our analyses on our behavioral outcome of interest. However, our supplemental specificity analyses suggested that the brain signatures and microbial profiles we found

in this study were not all linked specifically to internalizing symptoms in youth. The SOFA, MTL, SAL, PMN Network Connectivity Brain Signature, and Microbial Profile 1 seemed to capture contributions that were related to psychopathology generally, whereas the SOFA Between Network Connectivity Brain Signature and Microbial Profiles 2 and 3 seemed to capture contributions that were more specific to internalizing symptoms. As such, the symptom-anchored approach we used here had the intended benefit of identifying biological features that were relevant to our outcome of interest—internalizing symptoms, as well as in some cases to broader mental health outcomes. Beyond approaches that combine multi-dimensional biological datasets without reference to symptoms, incorporating a symptom-anchored approach can accelerate biomarker discovery by identifying symptom-relevant brain and microbial features from the outset, as opposed to identifying the most highly correlated/covarying features in general, which may or may not be symptom relevant. Another benefit of this analysis approach is that we can identify features with the strongest associations, followed by hypothesis testing, allowing for an exploratory investigation that includes the whole brain and gut microbiota community with inferential statistical tests. The prospective study design, in which we examined associations between the gut microbiota, brain, and internalizing symptoms across ages 2–7.5 years, allowed us to identify potential cascading effects of the early life gut microbiota on functional brain architecture, and on internalizing symptoms in middle childhood, an age of increased risk for internalizing symptoms[4,21]. Finally, our multi-ethnic sample of children living in Singapore contributes to diversity within gut microbiota research, which most often includes predominantly White samples living in North America and Western Europe.

Alongside the study's strengths, we note four important limitations. First, while our sample size was small ($N = 55$), our multivariate analysis method is robust to small sample sizes and a high number of predictors[35,37–39]. Nonetheless, there remains a small risk that the model was overfitted. Our findings should be replicated in larger samples with separate training and testing sets, and using complementary data analysis methods, to evaluate generalizability. Second, without repeated measures of the gut microbiota and RSFC, we were unable to test temporal ordering or change over time within the brain, gut microbiota, or internalizing symptoms. While the prospective study design allowed us to look at relations between the microbiota in early childhood and the brain in middle childhood, we don't know whether there were interventions (e.g., antibiotic exposure) that could have affected the gut microbiota in between our measurements. Thus, we speculate that the prospective associations we found might reflect programming effects or those that were resistant to intervention. Third, due to the high degree of compositional and functional diversity within a genus[87], we have limited insights into the possible functions that these microbes may perform. Future studies using higher resolution sequencing methods (i.e., to the species and strain level) and examining other "-omics" layers (e.g., transcriptomics, proteomics) can provide additional information about the specific functional processes within the early childhood gut microbiota that are related to brain development and internalizing symptoms in middle childhood. Finally, our sample comprises children from the community, which may limit the generalizability of the findings to clinical and other populations. Consistent with this community recruitment, the rate of cases at or above threshold for clinical levels of internalizing symptoms was 16% in this study, which aligns with epidemiological rates of internalizing symptoms in children[62,63].

In conclusion, we identified prospective associations between gut microbiota composition, brain functional connectivity networks, and internalizing symptoms across the early to middle childhood period. The analyses in this study using an exploratory multivariate approach followed by inferential hypothesis testing provide initial support for a

role of the early life gut microbiota in shaping mental health at school age via effects on functional brain development.

## Methods

### Participants and study design

Mother-child dyads participated in the GUSTO study, a large prospective mother-offspring cohort study. See Soh et al.[88] and Supplementary Methods for recruitment information. The GUSTO study was approved by the National Healthcare Group Domain Specific Review Board and SingHealth Centralized Institutional Review Board in Singapore. Mothers provided informed written consent for themselves and their children to participate.

The current study includes 110 children from the GUSTO cohort who donated a usable stool sample for microbiota analysis at age 2 years, contributed resting-state functional magnetic resonance imaging data at age 6 years, and whose caregiver reported on their internalizing symptoms at age 7.5 years, of which 55 had good-quality imaging data (see MRI Data Preprocessing in the Supplementary Methods for details). Child demographics and descriptive statistics on key study variables and covariates for the included and excluded samples are presented in Tables 1–3. We did not examine sex differences or sex interactions due to the small sample size and the lack of targeted hypotheses surrounding sex differences in our research questions.

Because the largest data loss occurred due to fMRI data availability and quality (of 318 individuals with stool samples and caregiver-reported internalizing symptom data, 110 had fMRI data, and only 55 had high-quality fMRI data), we report associations between the microbiota and caregiver-reported internalizing symptoms in the sample of $N = 318$ with these data in the SI.

### Measures

**Caregiver-reported internalizing symptoms: Child Behavior Checklist (CBCL).** We used parent-reported child symptoms at 7.5 years on the Child Behavior Checklist (CBCL)[89] as our primary indicator of child mental health. Parent-report was chosen over child self-report as clinician assessment tends to agree more with parent-report than child-report in the age range of this study[49]. The current study focused on the internalizing problems subscale (Cronbach's alpha = 0.81 in this sample), which is a composite of the anxious/depressed, withdrawn/depressed, and somatic complaints subscales. See Supplementary Methods and Supplementary Notes for symptom score preprocessing steps, potential covariates, variable missing data rates, and associations with child self-reported internalizing symptoms at a later time point.

### MRI data analysis

MRI data collection and preprocessing steps are described in the SI. Regions of interest (ROIs) were selected from the well-validated, functionally defined Seitzman 300-ROI parcellation atlas[90]. Each ROI is assigned to one of 13 predefined networks: auditory (AUD), cingulo-opercular (CON), default mode (DMN), dorsal attention (DAN), frontal parietal (FPN), MTL, parietomedial (PMN), SOFA, salience (SAL), dorsal somatomotor (SMD), lateral somatomotor (SML), ventral attention (VAN), and visual (VIS)[90]. For each participant, we extracted the time series of each ROI from the processed image and calculated RSFC strength using Fisher r-to-z transformation of the average correlation values between ROI-pairs within ($n = 13$; i.e., within-network connectivity) and between ($n = 78$; i.e., between network connectivity, e.g., DMN-FPN) each network. This approach has been used in previous studies examining functional network connectivity in youth[90–92].

### Gut microbiota bioinformatics

Gut microbiota collection, storage, DNA extraction, and sequencing steps are described in the SI. Reads were clustered into operational taxonomic units (OTUs) using USEARCH v9.2.64 at a 97% similarity threshold, and taxonomic classification was assigned to each OTU by comparing against the SILVA 123 ribosomal reference database. Microbiota data were normalized using rarefaction (depth cutoff = 5777) to account for uneven sequencing depth between samples (ten samples were removed due to having a sequencing depth shallower than the cutoff), and alpha diversity (within-individual bacterial community diversity) metrics were computed with QIIME v2.0[93]. Alpha diversity metrics included Observed Features (richness, or number of distinct taxa within the community), Pielou Evenness (relative evenness of taxa within the community), Shannon Index (richness weighted by evenness), and Faith's Phylogenetic Diversity (genetic diversity).

As is expected for 16S rRNA sequencing, most sequences were classified to the genus level. OTUs were aggregated to the genus level for downstream analysis. Where a genus-level classification was not available, the family-level classification was used instead. Genera abundance data were pre-processed using the following steps, which are recommended prior to multi-omics analysis[94]: (1) an offset of 1 was added to all abundances to allow for subsequent centered-log ratio transformation; (2) pre-filtering was performed to remove rare genera where the sum of abundances were below 5% of the total abundances in the dataset, which kept 64 out of the original 353 genera in the sample; (3) centered-log ratio transformation was applied to bring the data into unbounded space for downstream analysis.

### Data analysis plan

Analyses were conducted in R v4.2.3[95]. We followed a stepwise analysis approach whereby we started from the clinical outcome of interest (caregiver-reported internalizing symptoms at age 7.5 years) and worked backwards in time, first examining internalizing symptom associations with brain networks at age 6 years (brain signatures) using a multivariate approach, then examining derived brain signature associations with the gut microbiota at age 2 years, using both a univariate (alpha diversity) and a multivariate (microbial profiles) approach, and finally statistically testing whether the microbial profiles and alpha diversity indirectly related to caregiver-reported internalizing symptoms via the brain signatures. Each analysis step is described in further detail below (after the "Covariate selection" section).

**Covariate selection.** To be consistent with prior fMRI and gut microbiota psychopathology research in the GUSTO study cohort[44,96], covariates were selected for inclusion a priori. These included sex, gestational age, birthweight, mean framewise displacement (MRI), delivery mode, maternal education, and diet. While three indicators of diet were ultimately used as covariates—consumption of fiber, total fat, and polyunsaturated fat—there were in fact a total of eight diet indicators from which the final three indicators were selected: consumption of total fat, polyunsaturated fat, monounsaturated fat, saturated fat, fiber, carbohydrates, and protein in the diet, and age that the child stopped breastfeeding. To select covariates from the pool of eight diet indicators, we included those indicators that were significantly related to the microbiota, brain, and/or caregiver-reported internalizing symptoms. See Supplementary Methods for more details on the diet covariate selection process and significant associations.

Though the GUSTO study collected data on mothers' own mental health when children were aged 6 years, we chose not to control for maternal mental health in our analyses. While it is possible that mentally unwell parents may have a tendency to also report that their children are mentally unwell, correlations between parent self-reported and proxy-reported child mental health may also reflect the hereditary component of mental health[97], as well as influences of the child's and parent's shared environment[98]. In those conditions, controlling for maternal mental health may obscure real relationships with child mental health outcomes. Instead, we opted to test for the generalizability of associations in this study to other parent-proxy-reported child mental health

measures (e.g., externalizing symptoms) as well as to child self-reported depression symptoms later in development.

**Identification of brain networks that maximally covaried with internalizing symptoms—brain signatures.** We employed sPLS regression to identify linear combinations of brain networks at age 6 years (predictor dataset) that maximally covaried with caregiver-reported internalizing symptoms at age 7.5 years (outcome variable) using the *MixOmics* R package[99]. Briefly, sPLS regression is a multivariate integration technique similar to sparse canonical correlation analysis (sCCA) that achieves dimension reduction using lasso penalization and maximizes the covariance between orthogonal linear combinations of features from a multivariate dataset and an outcome variable(s) using PLS components[37,38]. sPLS differs from sCCA in that it seeks to maximize the covariance rather than the correlation, and assumes a predictor variable or dataset and an outcome variable or dataset. Thus, it is well-suited for instances in which a clear predictor and outcome relationship can be identified, for example, in this case, due to temporal ordering.

sPLS regression models were tuned for the optimal number of brain network components and variables per component using 10-fold cross-validation repeated 100 times. The model that maximized cross-validated $R^2$ was selected as the best-fitting. In the context of sPLS, $R^2$ is defined as the correlation between observed caregiver-reported internalizing symptoms and internalizing symptoms predicted by the model[100]. Outliers (third/first quartile $\pm 1.5 \times$ IQR) on any network were winsorized for that network, and model tuning was repeated on the winsorized dataset. Outlier identification details and other characteristics of participants flagged as outliers can be found in the SI. For each final tuned model, we report a Pearson's correlation between scores on the derived signature from the final model with the outcome variable, the loading of each network onto each component (which measures the relative strength of each network's contribution and the direction of its association with internalizing symptoms), the VIP of each network onto each component (which represents the percent of variance in caregiver-reported internalizing symptoms that is explained by the network divided by the total variance explained by the component), and the stability of each network (which represents how frequently the network was selected for inclusion across cross-validation repeats, or the reproducibility of the derived signatures when the training set is perturbed). We focus our interpretation on networks with stability ≥0.8, as is recommended[100].

Following tuning, derived latent component scores along with covariates were entered into multiple linear regression models predicting caregiver-reported internalizing symptoms, to test the statistical significance of the components' associations with internalizing symptoms and evaluate whether these relationships were robust to the inclusion of covariates.

**Identification of microbiota associations with brain signatures— microbial profiles and alpha diversity.** Next, we used sPLS regression, using the same methods as described in the previous analysis step, to identify linear combinations of genera abundances (i.e., microbial profiles) at age 2 years (predictor dataset) that maximally covaried with the latent component scores from the brain signatures (outcome variable) identified in the previous analysis step. Models were tuned and outliers were screened using the same procedures described in the SI. After microbiota component scores were derived, multiple linear regression models including latent microbiota component scores and covariates predicting latent component scores from each brain signature were run to evaluate statistical significance and robustness to covariate inclusion, as described in the previous analysis step.

To assess whether broad gut microbiota community compositional diversity was associated with the brain signatures, multiple linear regressions were run predicting each brain signature from each alpha diversity metric, controlling for covariates. Because we had 4 indicators of alpha diversity, statistically significant associations were subjected to multiple comparison correction using the Benjamin-Hochberg method[101], with a $q$ value threshold for significance of 0.25, as is recommended for biomarker discovery investigations[19].

**Mediation analysis: microbiota associations with caregiver-reported internalizing symptoms mediated by brain signatures.** Finally, we tested whether gut microbiota features (microbial profiles and alpha diversity metrics) were indirectly associated with caregiver-reported internalizing symptoms via the derived brain signatures with simple mediation models using the PROCESS macro in R[102] (Model 4), controlling for selected covariates. Total, indirect, and direct effects were estimated. The significance of indirect effects was determined using 95% confidence intervals with 5000 bootstraps.

### Supplemental analyses
We tested whether the microbiota and brain features that we identified to be predictive of caregiver-reported internalizing symptoms were driven by anxiety or depressive symptoms and/or also predictive of, and thus generalizable to, externalizing symptoms. We also tested whether these identified microbiota and brain features predicted child self-reported depressive or anxiety symptoms at 8.5 years of age. We also derived predicted microbiota functional profiles associated with each brain signature using predicted metagenomics from PiCRUST. Full methods and results for supplemental analyses are presented in the Supplementary Methods and Supplementary Notes.

### Inclusion and ethics statement
The research described in this manuscript has included local researchers (i.e., located in Singapore, the country where data collection was conducted) throughout the research process, including study design, implementation, data ownership, intellectual property, and publication authorship. The research is locally relevant, as determined by consultation with local partners. Roles and responsibilities were agreed among collaborators ahead of data analysis and write-up, but after data collection was already underway. The study was approved by local ethics review committees (National Healthcare Group Domain Specific Review Board and SingHealth Centralized Institutional Review Board). The authors have taken local and regional research relevant to the study into account in citations.

### Reporting summary
Further information on research design is available in the Nature Portfolio Reporting Summary linked to this article.

## Data availability
Data from the GUSTO study are not publicly available due to multi-institutional cohort data governance and ethical restrictions. Access procedures modeled after those of the NIH can be used to obtain access to the data. After a request is submitted, it is reviewed and approved by the GUSTO Executive Committee (of which C.Y.S., P.D.G., K.G. and M.J.M. are members). Applicants are required to comply with all applicable laws and partner with GUSTO investigators in order to access the data. Full collaboration guidelines can be found at: https://gustodatavault.sg/about/request-for-data. Requests can be sent to Li Ting (ang_li_ting@sics.a-star.edu.sg) to initiate the data access submission-to-approval workflow. Please allow 10 working days for a response. Data access is typically granted for 12 months. Source data are provided with this paper.

## Code availability
Scripts for data analysis steps in this manuscript are publicly available on GitHub: https://doi.org/10.5281/zenodo.16884691[103].

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

## Acknowledgements

The authors thank the GUSTO study team and Li Ting Ang in particular, for providing support with data access and preprocessing. The authors would also like to thank the GUSTO staff who collected and pre-processed the MRI, microbiota and symptom data. The authors also thank the Integrative Biostatistics and Bioinformatics Core at the UCLA Goodman Luskin Microbiome Center for their contributions to this study. Funding for the GUSTO study was provided by the Agency for Science Technology and Research (A*STAR), Singapore, under its Prenatal/Early Childhood Grant (Grant No. H22P0M0001) and the Singapore National Research Foundation under its Translational and Clinical Research Flagship Programme, administered by the Singapore Ministry of Health's National Medical Research Council, Singapore–NMRC/TCR/004-NUS/2008; NMRC/TCR/012-NUHS/2014. Additional funding was provided by the UK Medical Research Council (MC_UU_12011/4), the National Institute for Health and Care Research [NIHR Senior Investigator (NF-SI-0515-10042) and NIHR Southampton Biomedical Research Centre (NIHR203319)] and Alzheimer's Research UK (ARUK-PG2022A-008) to K.M.G.; the Singapore Institute for Clinical Sciences, Agency for Science Technology and Research, the JPB Foundation (US) Toxic Stress Network to M.J.M.; and the National Institute of Mental Health (NIMH) of the United States National Institutes of Health (NIH) under award numbers T32MH015750 to F.R.Q. and F32MH135657 to J.P.U. The content is solely the responsibility of the authors and does not necessarily represent the official views of the National Institutes of Health.

## Author contributions

F.R.Q., J.P.U., J.S.L. and B.L.C. contributed to the conception and design of the study. N.K., A.P.T., B.B.F.P.B., P.D.G., Y.S.C., H.C., M.V.F., L.M.D., F.Y., J.G.E., S.C., M.F.F.C., J.Y.T., K.M.G. and M.J.M. contributed to the acquisition of research funding, design of the parent GUSTO cohort study, and data collection. J.X. contributed to the processing of the gut microbiota data. F.R.Q. and J.P.U. contributed to the processing, analysis, and visualization of the data. F.R.Q., J.P.U., J.S.L. and B.L.C. contributed to the interpretation of results. F.R.Q., J.P.U. and B.L.C. wrote the initial draft. F.R.Q., J.P.U., J.S.L., B.L.C., B.B.F.P.B. and M.J.M. revised the manuscript, and all authors approved the manuscript.

## Competing interests

The authors declare no competing interests.

## Additional information

[1]Department of Psychology, University of California, Los Angeles, Los Angeles, CA, USA. [2]Department of Psychology, Stanford University, Stanford, CA, USA. [3]Vatche and Tamar Manoukian Division of Digestive Diseases, University of California, Los Angeles, Los Angeles, CA, USA. [4]G. Oppenheimer Center for Neurobiology of Stress and Resilience, Los Angeles, CA, USA. [5]Goodman-Luskin Microbiome Center, University of California, Los Angeles, CA, USA. [6]Department of Medicine, University of California, Los Angeles, Los Angeles, CA, USA. [7]Institute for Human Development and Potential, Agency for Science, Technology and Research in Singapore, Singapore City, Singapore. [8]Bioinformatics Institute, Agency for Science, Technology and Research in Singapore, Singapore City, Singapore. [9]Liggins Institute, University of Auckland, Auckland, New Zealand. [10]Department of Paediatrics, Yong Loo Lin School of Medicine, National University of Singapore, Singapore City, Singapore. [11]Department of Obstetrics and Gynaecology, National University Health System, Singapore City, Singapore. [12]Psychiatry and Radiology, KK Women's and Children's Hospital, Singapore City, Singapore. [13]Duke-National University of Singapore Medical School, Singapore City, Singapore. [14]Department of Child Development, KK Women's and Children's Hospital, Singapore City, Singapore. [15]Department of Paediatrics, KK Women's and Children's Hospital, Singapore City, Singapore. [16]Department of Pediatrics, Lee Kong Chian School of Medicine, Nanyang Technological University, Singapore City, Singapore. [17]Department of Maternal Fetal Medicine, KK Women's and Children's Hospital, Singapore City, Singapore. [18]Department of Obstetrics and Gynaecology, Yong Loo School of Medicine, National University of Singapore, Singapore City, Singapore. [19]Department of General Practice and Primary Health, University of Helsinki and Helsinki University Hospital, Helsinki, Finland. [20]Program of Public Health Research, Folkhälsan Research Center, Helsinki, Finland. [21]Saw Swee Hock School of Public Health, National University of Singapore, Singapore City, Singapore. [22]Medical Research Council Lifecourse Epidemiology Centre, University of Southampton, Southampton, UK. [23]NIHR Southampton Biomedical Research Centre, University Hospital Southampton NHS Foundation Trust and University of Southampton, Southampton, UK. [24]Department of Psychiatry, Douglas Hospital Research Centre, McGill University, Montreal, QC, Canada. [25]Brain–Body Initiative, Agency for Science, Technology and Research, Singapore City, Singapore. [26]These authors contributed equally: Francesca R. Querdasi, Jessica P. Uy. ✉e-mail: frq@ucla.edu

