## [Transparent Peer Review file · Nature Communications]

Childhood gut microbiome is linked to internalizing symptoms at school age via the functional connectome

Corresponding Author: Ms Francesca Querdasi

Version 0:

Reviewer comments:

Reviewer #1

(Remarks to the Author)

This paper examines how early gut microbiome composition is linked to later functional brain networks and mental health development. I commend the authors on their innovative and theoretically motivated approach to variable selection and dimension reduction. However, there are several issues with the manuscript.

My primary concern lies in the sample and sample size ($N = 55$). Although the authors write about how the analyses are appropriate for the sample size in the discussion section there are no citations listed for this claim. I understand that further data collection is unrealistic but I do wonder if the majority of the data loss is for the fMRI if the researchers can run additional analyses in participants that have usable stool samples and CBCL data (even though findings would suggest minimal direct associations).

I also had several questions related to internalizing symptoms. The authors mention using CBCL raw scores but the mean and SD in the table look to be in the range of t-scores. Please clarify this. The authors also chose to do a transformation – here it would be helpful to plot the distributions of data before and after the transformation. Finally, this seems to be a relatively healthy sample. It would be helpful if the authors listed the number of children that meet the clinical criteria. If this is a limited number – this would need further discussion about what clinical conclusions can be reached if they are examining symptom levels in a low to medium-symptom cohort. I also wonder why the authors only examined the internalizing symptoms and did not examine individual subscales (e.g., anxiety, depression) or even look at externalizing symptoms. I think the further clinical characterization of the sample would add to the paper. It is also important to remember an important limitation of the CBCL is that it is a parent report measure – and having a covariate to help account for the parent's own mental health (e.g., maternal depressive symptoms) can help to alleviate reporter-driven biases.

It is interesting that so many between network connections had high values of variable importance. I think a bit more review and reflection of why within and between network connections are important to the gut and mental health would be of help in the introduction and discussion.

Given the previous work using this dataset (Querdasi et al., 2023) – I think it is important to highlight what the present study adds beyond what has already been found.

It would be helpful to quantify how many children had the genera of interest for each of the clusters identified. In addition to a descriptive figure of the average abundances of each of the genera identified for the sample. Also, some work has examined associations at the genera level whereas other work is at the species level. Further discussion of the potential impacts of that decision would be helpful.

It would be helpful to test for differences in socio-demographics and variables of interest for participants that are included vs excluded from the sample. In addition to including further investigation into the participants flagged as outliers.

Minor comments –

Figure S1. Add labels to the other side of the correlation matrix

Figure S2 and Table S3 can be combined.

Some of the network acronyms may not have been previously defined.

(Remarks on code availability)

The code provides a nicely formatted R script. It would be helpful to have additional data describing the format of each of the data frames or even a blank dataframe with variable labels to further understand the structure of inputs.

Reviewer #2

(Remarks to the Author)

This is an interesting study examining links between the child gut microbiome, brain networks, and internalizing symptoms later in childhood. There are several areas that could use some clarification or elaboration:

1. The manuscript may overstate the novelty of the tested associations. There is a robust body of research testing associations between microbiome, brain, and behavior early in development (including resting-state functional connectivity). The novelty is more evident in the longitudinal nature of their data and use of functional networks for brain outcomes.
2. Early childhood is a vast period with many different developmental periods that play a crucial role in how experiences are processed and how they affect the developing body and brain. It would help to add more details about the developmental periods in the review of existing literature. For example, on p. 4 there are references to several studies "... microbiome composition is associated with neurocognitive and behavioral outcomes during childhood, such as internalizing symptoms" ; "... microbiome composition and differential abundance of single microbes, in infancy and toddlerhood" and "studies of children show that the functional neural connectome..". Stating the specific age ranges of the children in these studies, as well as whether they were cross-sectional or longitudinal, would provide necessary information to determine if the current investigation (and the developmental periods chosen) is an appropriate and novel next step.
3. Better justification is needed for why resting-state functional data is relevant for clinical outcomes and the gut microbiome.
4. Neuroimaging at this age is difficult, and only 55 of 110 had useable imaging data. More details about missing data would help to evaluate whether this missingness was random or related to primary variables – ie., were there differences in child characteristics between those with useable data vs not? For example, are internalizing children more compliant and follow directions better than externalizing (or vice versa, internalizing children may be more nervous about being in the scanner). This may lead to bias with internalizing children being over or underrepresented in the sample.
5. It would be helpful to assess the model fit of the mediation analyses. Since the authors are testing multiple gut microbiome metrics as predictors of the pathway, evaluating model fit could provide insight into how well the mediation model captures the overall data patterns. Given the complexity of the associations between the gut microbiome and the brain, poor model fit—regardless of statistical significance—could indicate issues with the magnitude of the observed effects.
6. Although it's hard to avoid, the use of CBCL as the only outcome variable is not ideal. Parent-report of child behavior comes with clear limitations (subjective perceptions, bias, embarrassment) and does not reflect clinical significance. This should be included as a limitation – and perhaps consider using the phrase "parent reported internalizing symptoms" throughout to be clear that this is not a clinical diagnosis.
7. A major concern with the covariate selection is that the authors primarily chose covariates based on prior fMRI and microbiome literature. However, for the diet-related covariates, they used a data-driven approach, which seems contradictory. For example, several potential covariates—such as maternal age, breastfeeding, or formula feeding—could have yielded significant results outside of the diet analysis if tested. The covariate selection process should be consistent throughout.
8. Authors should do multiple comparison correction in their regression analyses.

(Remarks on code availability)

N/A - I am not the appropriate person to review/evaluate this code.

Reviewer #3

(Remarks to the Author)

(Remarks on code availability)

Reviewer #4

(Remarks to the Author)

The manuscript "Childhood gut microbiome is linked to mental health at school age via the functional connectome" describes links between gut microbita at 2 years, connectome at 6 years, and internalizing symptoms at 7.5 years.

While combining multiple data types is very interesting and valuable, I am afraid that the models are overfitted due to the low sample size. Could you test with some other method(s) if the results and interpretations are robust?

Moreover, better inferences on the functional potential or output of the gut microbiome would be more informative. The authors do acknowledge this in the discussion, and if gaining new data - such as faecal metagenome or serum

metabolomics or so forth - is not feasible, I would recommend predicting functional pathways based on the amplicon sequencing data. It often helps with the inference, and you could even compute sPLS similarly of the predicted metagenome.

Thirdly, the authors would have access to wider array of psychiatric symptom domains, and I would believe that there would be many relevant outcomes related to gut-brain axis. Could the authors provide better rationale on why you particularly focused on internalizing symptoms and not e.g. inattention?

Minor comments:

Since the manuscript contains amplicon sequencing data, the correct term would be microbiota, not microbiome.

The introduction seems to be a little cumbersome to read, and I wonder whether you could focus a little less on the different model options, since this is not a methodological paper anyway?

I would repeat the sample size (with and without covariates) and description of the sample in the beginning of the results. I would like to read on the variation of symptoms, inclusion and exclusion criteria, and information on key confounders, such as antibiotic and other medication use, psychiatric diagnoses, and if there were any high-risk families (SES, income, parental psychiatric diagnoses) and so forth.

In addition, since the results are before methods, please help the reader and list the covariates used in the analyses.

It is also helpful for a non-expert reader to open the abbreviations when first mentioned.

Since there is no repeated assessment of microbiome, psychiatric symptoms or connectome, I believe the correct term is a cross-sectional study (in a longitudinal cohort study though).

Can you explain in brief the gut microbiota methodology in the main text as well? In addition it is great to know if you estimated ASVs, the taxonomy data based and so forth. I would also report for instance in the supplements if you used any controls.

In the supplements, I cannot really parse the sentence "We used alpha diversity as this microbiome feature was not derived from the brain or internalizing symptoms."

I commend the authors for sharing the analytical pipeline. I would also recommend to use the STORMS checklist.

- Anna Aatsinki

(Remarks on code availability)

Reviewer #5

(Remarks to the Author)

The study investigates the influence of infant microbiome composition on later brain and behavioral development using a sequential sparse Partial Least Squares (sPLS) approach. The authors identify key microbial signatures associated with internalizing symptoms, mediated through brain networks. I have several concerns regarding its design and methodology. In the introduction, the authors should provide additional context regarding the stability and development of the microbiome from ages 2 to 8, as well as the developmental trajectory of the connectivity measures used. This would help readers better understand the rationale behind examining microbiome composition at 2 years in relation to resting-state functional connectivity (RSFC) and behavioral outcomes measured at 6–7.5 years.

In the covariates, the authors do not mention any SES variables. Why was that not included given its role in affecting all three domains the authors tested.

The use of a sequential sPLS approach with a relatively small sample size ($n = 55$) raises concerns about the robustness of the findings. Was stability selection implemented to ensure that the selected features are consistent across bootstrapped models? Without such measures, there is a significant risk of overfitting as mentioned in their limitations. Additionally, selecting the best-fitting model based solely on R^2 in a high-dimensional dataset with a limited sample size is concerning, as it may lead to R^2 inflation due to overfitting. I strongly recommend reporting cross-validated R^2 or adjusted R^2 to provide a more reliable measure of model performance.

Furthermore, I have concerns of the use of sequential sPLS modeling approach. Given the small sample size, errors or biases introduced in the first sPLS step (linking neuroimaging data to CBCL scores) are likely to propagate to the subsequent step. Would an integrative approach, such as DIABLO, which analyzes all datasets simultaneously, be more appropriate? I encourage the authors to explore this alternative and compare the results.

Regarding the selection of features for the second sPLS step, the authors report using the features with the highest loadings

on the first two components. Was the latent score used as an input for the next sPLS step? If not, I would like to see those results included for comparison.

Additionally, in the discussion, the authors should elaborate on the relevance of their findings. Specifically, they need to clarify the rationale for how microbiome composition at 2 years of age could influence brain connectivity measures at 6 years. Providing a justification for the chosen time points would help with the interpretation of the results and contextualize their implications.

Overall, while this study presents an interesting contribution to our knowledge on microbiome-brain-behavior interactions, addressing these methodological concerns would strengthen the validity and interpretability of the findings.

(Remarks on code availability)

There is sufficient information provided in the Github page to reproduce the analysis steps.

Version 1:

Reviewer comments:

Reviewer #1

(Remarks to the Author)

The authors have thoroughly addressed my comments.

(Remarks on code availability)

The code provides useful insights into the data analysis conducted for this research

Reviewer #2

(Remarks to the Author)

Thank you for the opportunity to review the resubmission of the manuscript "Childhood gut microbiome is linked to mental health at school age via the functional connectome" which examines links between gut microbiota at 2 years, connectome at 6 years, and internalizing symptoms at 7.5 years. The authors have done a very thoughtful job at responding to my initial concerns. The manuscript now does a better job at clarifying and justifying the use of resting state functional data, reviewing literature across developmental periods, and explaining the decisions made around multiple comparisons and model fit. I believe my comments were all addressed adequately.

(Remarks on code availability)

Reviewer #3

(Remarks to the Author)

(Remarks on code availability)

Reviewer #4

(Remarks to the Author)

This is the second time I am reviewing the manuscript "Childhood gut microbiome is linked to internalizing symptoms at school age via the functional connectome."

I appreciate that the authors have addressed most of my previous suggestions. However, I was left wondering why some of the supplemental analyses and results are reported only in the Methods section and the Supplementary Information (SI). I recommend moving the description of the predicted functional pathways to the Results section to ensure it does not escape readers' attention. The authors have explained this aspect in more detail in their response letter.

Regrettably, there is little that can be done about the low sample size, and having specified hypothesis as presented here definitely is a merit. However, the low sample size may lead to difficulty in testing whether the current interpretations would hold if alternative, complementary data analysis methods were applied.

(Remarks on code availability)

Reviewer #5

(Remarks to the Author)

The authors have adequately addressed my questions and concerns.

(Remarks on code availability)

Reviewer #1:

This paper examines how early gut microbiome composition is linked to later functional brain networks and mental health development. I commend the authors on their innovative and theoretically motivated approach to variable selection and dimension reduction. However, there are several issues with the manuscript.

We thank the reviewer for their positive comments about the manuscript. We have addressed their concerns below.

My primary concern lies in the sample and sample size (N = 55). Although the authors write about how the analyses are appropriate for the sample size in the discussion section there are no citations listed for this claim. I understand that further data collection is unrealistic but I do wonder if the majority of the data loss is for the fMRI if the researchers can run additional analyses in participants that have usable stool samples and CBCL data (even though findings would suggest minimal direct associations).

We regret that we did not initially include citations for our claims about our analyses (sPLS) being appropriate for our sample size. We have rectified this omission in the revised manuscript:

Starting Line 175: *“Multivariate machine learning approaches address key limitations of univariate neuroimaging analyses, including small effect sizes, and the inability to identify multivariate combinations of brain patterns that relate to psychopathology.³⁴ Among these, canonical correlation analysis (CCA) and partial least squares (PLS) are widely used to reduce complex datasets into latent components that maximize shared variance–correlation (CCA) or covariance (PLS).^{35,36} PLS is particularly advantageous in situations when the sample size is similar to, or much smaller than, the number of variables in the dataset, and in situations where multicollinearity is present.^{35,37,38} Incorporating sparsity into the PLS can facilitate interpretation and reduce overfitting by selecting a subset of relevant variables,^{35,37,38} making this approach (sparse PLS, or sPLS) especially robust to small sample sizes.³⁹”*

Starting Line 185: *“While multivariate machine learning techniques, including sPLS, are frequently used to compress microbiota data and identify its associations to other variables—including within a multi-omics framework,⁴⁰ clinical outcomes,⁴¹ and with the brain—^{42,43} these techniques have been underused in developing populations, which are often characterized by the small samples for which these approaches are particularly well suited.”*

Moreover, we now include citations to our claim in the discussion, starting Line 557: *“First, while our sample size was small (N=55), our multivariate analysis method is robust to small sample sizes and a high number of predictors.^{35,37–39} Nonetheless, there remains a small risk that the model was overfitted and our findings should be replicated in larger samples with separate training and testing sets to evaluate generalizability.”*

As suggested by this reviewer, we also ran additional analyses on the larger sample of participants who had usable stool samples and CBCL data (N = 318). We calculated microbial

profile scores for these participants using the equations that were derived in the analytic sample, and then tested the scores' associations with parent-reported internalizing symptoms in youth. Results were similar to those in the analytic subsample (and are reported in the SI): Scores on all microbial profiles were not significantly associated with parent-reported internalizing symptoms, controlling for covariates. This was indeed expected because our original findings were that there were no total or direct effects of microbial profile scores on parent-reported internalizing symptoms, and validates that the analytic sample is a reflection of the wider cohort.

I also had several questions related to internalizing symptoms. The authors mention using CBCL raw scores but the mean and SD in the table look to be in the range of t-scores. Please clarify this.

We did in fact use the raw scores in our analyses, as per recommendations for research use from the CBCL (Achenbach & Rescorla, 2001). Given that analytical choice, we agree that it is confusing for us to list the T-score descriptives in the table. We have now changed the mean, SD, and range in Table 1 to be the raw scores, and we also report the percentage of children who scored above the clinical cutoff based on t-scores in Table 1 (relevant rows reproduced below).

Table 1. Demographics and descriptive statistics: Mean (SD); Range for continuous variables and N (%) for categorical variables.

Measure	Analytic Sample (N=55)	Participants Excluded in fMRI QC (N=55)	Significant difference between analytic and excluded sample
Internalizing symptoms (raw scores)	5.44 (4.65); 0-19	6.09 (6.24); 0-29	No
Clinical symptom levels (T-score > 63)	6 (11%)	7 (13%)	
Borderline symptom levels (T-score: 60-63)	3 (5%)	5 (9%)	
Not clinically significant symptom levels (T-score < 60)	46 (84%)	43 (78%)	

The authors also chose to do a transformation – here it would be helpful to plot the distributions of data before and after the transformation.

We have added a figure to the supplement (Figure S2, reproduced below) that shows the distribution of internalizing symptoms before and after the transformation.

Figure S2. Distribution of caregiver-reported internalizing symptom raw scores before (A, purple) and after (B, green) Box-Cox transformation.

Finally, this seems to be a relatively healthy sample. It would be helpful if the authors listed the number of children that meet the clinical criteria. If this is a limited number – this would need further discussion about what clinical conclusions can be reached if they are examining symptom levels in a low to medium-symptom cohort.

We now report the number of children meeting clinical criteria in Table 1: 11% with clinical levels, 5% with borderline clinical levels, and 84% without clinically significant levels.

We also include a brief discussion of this outcome and what clinical conclusions can be reached in a low to medium-symptom cohort in the discussion section (page 11):

Starting Line 439: “Critical to the interpretation of internalizing symptom associated brain signatures, in the current study, only a minority of children exhibited symptoms at clinical or borderline levels, consistent with prevalence rates in the general population.^{62,63} However, because symptom severity exists along a continuum, focusing exclusively on clinically significant distress (i.e., scores above diagnostic cutoffs) may overlook meaningful variation in functioning. Subclinical symptoms, particularly of depression, have been shown to increase the risk of later diagnoses of major depressive disorder—potentially through pathways involving cognitive dysfunction and anhedonia.^{64–66} Given the younger age of this sample, which precedes the median age of onset for disorders such as anxiety (11 years) and depression (30 years),

examining subthreshold symptoms may offer valuable insights into early indicators of risk and future outcomes.⁶⁷

We also include a discussion of the limitations of using a community sample in the limitations section of the discussion (page 14, starting line 573):

“Finally, our sample comprises children from the community, which may limit the generalizability of the findings to clinical and other populations. Consistent with this community recruitment, the rate of cases at or above threshold for clinical levels of internalizing symptoms was 16% in this study, which aligns with epidemiological rates of internalizing symptoms in children.^{62,63}”

I also wonder why the authors only examined the internalizing symptoms and did not examine individual subscales (e.g., anxiety, depression) or even look at externalizing symptoms. I think the further clinical characterization of the sample would add to the paper.

While we acknowledge that the microbiome-gut-brain-axis has been implicated in a range of mental health outcomes (including both internalizing and externalizing symptoms), the scope of the current investigation focused on identifying specific microbiota and functional network-level predictors of childhood internalizing symptoms for two reasons. First, internalizing symptoms during early-middle childhood increases risk for chronic and recurrent internalizing symptoms across the lifecourse making this a particularly important symptom group to investigate for links to the microbiome. Second, recent evidence suggests that the associations between microbiota composition (alpha diversity and specific microbes) are more robustly associated with internalizing symptoms and somatic complaints than externalizing symptoms during the preschool years (van de Wouw et al., 2022).

These points have now been added to the introduction section of the manuscript (page 6, starting line 209):

“We focus on internalizing problems for two reasons: First, internalizing symptoms during early-middle childhood increases risk for chronic and recurrent internalizing symptoms across the lifecourse,^{45,46} making this a particularly important symptom group to investigate for links to the microbiota. Second, recent evidence suggests that the associations between microbiota composition (alpha diversity and specific microbes) are more robustly associated with internalizing symptoms and somatic complaints than externalizing symptoms during the preschool years.¹⁴”

Nonetheless, while we decided to retain our focus on internalizing symptoms for this paper, we added analyses to the supplement that tested whether the microbiota and brain features that we identified to be predictive of internalizing symptoms were (1) driven by anxiety or depressive symptoms and/or (2) also predictive of, and thus generalizable to, externalizing symptoms, and child-self-reported depressive symptoms. We found that the SOFA, MTL, SAL, PMN Network Connectivity Brain Signature was positively associated with caregiver-reported depression, anxiety, and externalizing symptoms at age 7.5 years, as well as with child self-reported

depressive symptoms reported at age 8.5 years, whereas the SOFA Between Network Connectivity Brain Signature was associated with child self-reported depressive symptoms reported at age 8.5 years. In terms of the microbial profile's links with child outcomes, we observed that Microbial Profile 1 (defined through association with the SOFA, MTL, SAL, PMN Network Connectivity Brain Signature) was positively associated with caregiver-reported externalizing symptoms. In contrast, Microbial Profiles 2 and 3 (both defined through association with the SOFA Between Network Connectivity Brain Signature) were not associated with any additional symptom measures (results reported in the SI). Though most of these associations were not robust to the inclusion of covariates, they suggest that the SOFA, MTL, SAL, PMN Network Connectivity Brain Signature and associated Microbial Profile 1 may capture contributions that are related to psychopathology generally, whereas the SOFA Between Network Connectivity Brain Signature and associated Microbial Profiles 2 and 3 may capture contributions that are more specific to internalizing symptoms.

Starting Line 375: *“Specificity of Brain Signatures and Microbiota Profiles to Caregiver Reported Internalizing Symptoms*

To test specificity of links between Brain Signatures and Microbial Profiles to caregiver-reported internalizing symptoms broadly, we examined all identified Brain Signatures and Microbial Profiles and their links with caregiver reported anxiety and depressive symptoms separately at 7.5 years of age (which go into the internalizing symptom score), as well as with caregiver-reported externalizing symptoms at 7.5 years of age, and with child self-reported depression symptoms at 8.5 years of age. We found that the SOFA, MTL, SAL, PMN Network Connectivity Brain Signature was positively associated with caregiver-reported depression, anxiety, and externalizing symptoms at 7.5 years of age, as well as with child self-reported depressive symptoms at 8.5 years of age. In other words, this Brain Signature was not specific to the caregiver-reported internalizing symptoms upon which it was defined. In contrast, the SOFA Between Network Connectivity Brain Signature was associated with child self-reported depressive symptoms at 8.5 years of age, suggesting it too was not completely specific to the caregiver-reported internalizing symptoms upon which it was defined. Similarly, Microbial Profile 1 was positively associated with caregiver-reported externalizing symptoms at 8.5 years of age, again suggesting some degree of generalizability beyond caregiver-reported internalizing symptoms. In contrast, Microbial Profiles 2 and 3 were not associated with any additional symptom measures, which suggests that the indirect association between Microbial Profile 3 and caregiver-reported internalizing symptoms reported above (mediated by the SOFA Between Network Connectivity Brain Signature) is indeed specific to that outcome measure. Methods and full results are presented in the SI.”

Starting Line 533: *“However, our supplemental specificity analyses suggested that the brain signatures and microbial profiles we found in this study were not all linked specifically to internalizing symptoms in youth. The SOFA, MTL, SAL, PMN Network Connectivity Brain Signature and Microbial Profile 1 seemed to capture contributions that were related to psychopathology generally, whereas the SOFA Between Network Connectivity Brain Signature and Microbial Profiles 2 and 3 seemed to capture contributions that were more specific to internalizing symptoms. As such, the symptom anchored approach we used here had the intended benefit of identifying biological features that were relevant to our outcome of interest - internalizing symptoms, as well as in some cases to broader mental health outcomes. Beyond*

approaches that combine multidimensional biological datasets without reference to symptoms, incorporating a symptom anchored approach can accelerate biomarker discovery by identifying symptom-relevant brain and microbial features from the outset, as opposed to identifying the most highly correlated/covarying features in general, which may or may not be symptom relevant.”

To add to the clinical characterization of the sample, we also now include descriptive statistics on caregiver-reported youth externalizing, depression, and anxiety symptoms in Table 1, as well as the percentage of the sample exhibiting clinically significant levels of caregiver-reported internalizing and externalizing symptoms based on CBCL t-score clinical cutoffs.

It is also important to remember an important limitation of the CBCL is that it is a parent report measure – and having a covariate to help account for the parent’s own mental health (e.g., maternal depressive symptoms) can help to alleviate reporter-driven biases.

There are several reasons that we trust the parent-reported symptom measure from the CBCL and use it as our primary indicator of child mental health. First, in this age group, clinician assessment tends to agree more with parent-report than child-report (Hawley & Weisz, 2003). Moreover, discrepancies between parent reported and child self-reported mental health symptoms may not always be larger for parents with more psychopathology; in fact, some work has found they are greater for parents with *less* psychopathology (Booth et al., 2023). We now bring these points up in the methods section of the paper:

Starting Line 608: *“We used parent-reported child symptoms at 7.5 years on the Child Behavior Checklist (CBCL)⁸⁹ as our primary indicator of child mental health. Parent-report was chosen over child self-report as clinician assessment tends to agree more with parent-report than child-report in the age range of this study.⁴”*

To further explore and deal with the issue of potential parent-reporting bias in our sample, we have now conducted supplemental analyses examining associations between parent-reported internalizing symptoms at child age 7.5 years and child self-reported symptoms at age 8.5 years, and found that they are significantly positively correlated. Specifically, we find that the CBCL internalizing symptoms subscale is positively correlated with child self-reported depressive symptoms ($r = .41$, $t(31) = 2.47$, $p = .02$) and marginally correlated with child self-reported anxiety symptoms ($r = 0.30$, $t(31) = 1.72$, $p = .095$) at 8.5 years of age. We have included these analyses in the supplement accompanying the main text.

Because a significant correlation between parents' report of their own and their child's mental health could reflect several factors in addition to bias, including the hereditary component of mental health (Jami et al., 2022) and influences of the child’s and parent’s shared environment (Burt, 2009), we have decided to not control for maternal mental health in our analyses. We include a brief discussion of this point in the methods.

Starting Line 676: *“Though the GUSTO study collected data on mothers’ own mental health when children were aged 6 years, we chose not to control for maternal mental health in our analyses. While it is possible that mentally unwell parents may have a tendency to also report*

that their children are mentally unwell, correlations between parent self-reported and proxy-reported child mental health may also reflect the hereditary component of mental health,⁹⁷ as well as influences of the child's and parent's shared environment.⁹⁸ In those conditions, controlling for maternal mental health may obscure real relationships with child mental health outcomes. Instead, we opted to test for generalizability of associations in this study to other parent-proxy reported child mental health measures (e.g., externalizing symptoms) as well as to child self-reported depression symptoms later in development.”

It is interesting that so many between network connections had high values of variable importance. I think a bit more review and reflection of why within and between network connections are important to the gut and mental health would be of help in the introduction and discussion.

Thank you for this important comment. We agree that the importance of the within and between network connections was missing from the introduction and we have remedied that in the revised manuscript. The developing brain typically follows a trajectory of increasing within-network connectivity and increasing between-network anti-correlation or segregation, reflecting increasing specialization of network-related functions and more efficient information processing (Gao et al. 2015; Fransson et al. 2009; Fair et al. 2008; Sherman et al., 2014; Rebello et al., 2018; Fan et al., 2021). In the context of internalizing symptoms, for example, greater between-network connectivity between the default mode network (DMN) and networks related to attention and salience detection may reflect the internal orientation of emotionally salient stimuli (e.g., rumination) (Tang et al., 2018; Webb et al., 2021).

To contextualize the current study, we now include in the introduction a review of extant microbiota-brain studies in developing populations, a brief description of gut microbiota development, some of the pathways through which the gut microbiota shape brain development, the developmental trajectories of resting state network connectivity, and their implications for psychopathology in youth—Lines 119–173:

“Recent evidence that microbiome composition is associated with neurocognitive and behavioral outcomes during childhood, such as internalizing symptoms,^{13–15} underscores the importance of understanding the MGBA's contribution to mental health during development. To date, however, most studies examining the MGBA in the context of typical development have focused on infancy and toddlerhood, with an emphasis on neurocognitive, rather than mental health, outcomes.^{16–19} These studies have found that individual differences in microbial alpha diversity are associated with variation in the structure and function of brain regions and networks implicated in sensorimotor processing, language, and emotion.^{16–19} For example, greater alpha diversity in infancy has been concurrently linked to increased fronto-parietal connectivity—a network implicated in cognitive control, which in turn was associated with increased infant negative emotionality.¹⁹ Similarly, higher alpha diversity at age 1 year has been associated with lower functional connectivity between the left amygdala and thalamus, and between the anterior cingulate cortex and insula, as well as greater connectivity between the supplemental motor area and inferior parietal lobule.¹⁸ Higher alpha diversity at age 1 year has also been linked with poorer cognitive performance (visual reception and expression language) and brain volume at later developmental stages—greater left precentral, right angular, and left

amygdala volumes at age 2 years¹⁶—highlighting potential programming effects of early microbiota on later brain structure.

In studies examining specific microbial taxa, concurrent and prospective associations with neurodevelopment have also been reported. For instance, Carlson et al.¹⁷ found that relative abundances of Streptococcus and Staphylococcus at age 1 month were concurrently negatively associated with amygdala volumes, and positively associated with medial prefrontal cortex (mPFC) volumes. In terms of potential programming effects, higher abundance of Bacteroides at age 1 month was linked to smaller mPFC at age 1 year, whereas greater Lachnospiraceae abundance was prospectively associated with larger amygdala volume - an association that was reversed for Enterobacteriaceae.¹⁷ Notably, in that study, there were no concurrent associations between specific bacterial taxa and brain volumes at age 1 year, suggesting cascading developmental effects, whereby earlier microbial composition influences later brain structure. In that study, concurrent and prospective associations between microbes and affective outcomes were also reported: higher abundances of Dialister, and bacteria that are part of the Clostridiales order, at 1 year were associated with reduced fear behavior at 1 year, and alpha diversity at 1 month was associated with increased fear behavior at 1 year. These findings align with the hypothesis, also supported by research in rodents,²⁰ that early-life microbiota provide critical signaling inputs necessary for typical neurodevelopment, and that disruptions in these signals may confer risk for later psychopathology.⁶ Despite these insights, it remains unclear whether—and how—early life microbial diversity and composition affect brain function later in development, during middle childhood, a high risk period for mental health problems.^{4,21}

Of the developmental studies linking the microbiota to brain function, most have employed seed-based resting state functional connectivity (RSFC) analyses.^{16,18} While use of RSFC is preferable in developing populations due to reduced task demands, the use of researcher-selected seeds in these analyses provide a limited view of brain function, and further, does not account for connectivity within and between canonical large-scale functional brain networks, which have been linked to neurocognitive behaviors and psychopathological outcomes in children.^{22–28} For example, internalizing symptoms in children have been associated with alterations in RSFC within the default mode network (DMN), and the ventral attention network (VAN), and between both DMN and striatal regions with cingulo-opercular (CON) networks.²² Across development, a pattern of increasing within-network integration and between-network segregation supports the increasing specialization of network-related functions and efficient information processing—deviations from which are associated with psychopathology in youth.^{29–33} As such, it is clear that investigating the impact of early microbiota on the development of brain network integration and segregation is an essential step for understanding the microbiota's contributions to psychopathology. However, such analyses are challenging due to the multivariate complexity of both microbiota and large-scale brain network data. Use of multivariate machine learning analysis techniques may help to solve this issue.”

We have kept the introduction focused on interpreting the main findings of the paper and do not reiterate this information about within and between network connectivity there.

Given the previous work using this dataset (Querdasi et al., 2023) – I think it is important to highlight what the present study adds beyond what has already been found.

We thank the reviewer for this suggestion. We now include in the introduction what the current study adds beyond prior work, starting Line 202:

“In previous analyses of data from this cohort, we demonstrated that maternal childhood maltreatment, maternal prenatal anxiety, and children’s exposure to stressful life events were associated with distinct gastrointestinal microbiota profiles at 2 years of age, which were in turn associated with behavioral problems at 2 and 4 years of age.⁴⁴ In the current study, we investigate how gastrointestinal microbiota composition at 2 years of age relates to children’s intrinsic functional brain network connectivity at 6 years of age, with these brain networks defined by their association with caregiver-reported internalizing problems at 7.5 years of age. We focus on internalizing problems for two reasons: First, internalizing symptoms during early-middle childhood increases risk for chronic and recurrent internalizing symptoms across the lifecourse,^{45,46} making this a particularly important symptom group to investigate for links to the microbiota. Second, recent evidence suggests that the associations between microbiota composition (alpha diversity and specific microbes) are more robustly associated with internalizing symptoms and somatic complaints than externalizing symptoms during the preschool years.¹⁴ This multivariate symptom-centered approach to identify specific brain functional architecture relevant to mental health, before linking it to the microbiome is a novel approach that can provide potential targets for future interventions. Furthermore, our examination of prospective links between early microbiota and brain networks/behavioral outcomes in middle childhood provides insight into the potentially long reach of early microbial composition on child outcomes, especially during periods of risk for the emergence of internalizing symptoms.^{47,48}”

It would be helpful to quantify how many children had the genera of interest for each of the clusters identified. In addition to a descriptive figure of the average abundances of each of the genera identified for the sample.

We have added a column “N not zero” to Table S3, the table that includes loadings, VIP, and stability for each genus selected for inclusion in the Microbial Profiles.

We have added a descriptive figure (Figure S3) that visualizes the distribution of relative abundance, and average relative abundance, for each genus selected for inclusion in the Microbial Profiles.

Also, some work has examined associations at the genera level whereas other work is at the species level. Further discussion of the potential impacts of that decision would be helpful.

We examine associations at the genus level, rather than species level, because the microbiome sequencing method (16S rRNA amplicon sequencing) used in our study is inappropriate for examining composition at the species level—it is not reliable for species- or strain-level resolution (see (Janda & Abbott, 2007) for a review). 16S rRNA sequencing has limited ability to provide species-level identification, in part because it cannot distinguish between phylogenetically similar species (i.e., those that have recently diverged from each other on the phylogenetic tree). On the other hand, 16S rRNA sequencing performs well for broader, genus-level identification (classification is possible for >90% of microbes) and exploring community diversity. Thus, we

chose to report results at the genus level, because it is the most specific level with good performance for our sequencing method. We discuss this limitation of 16s rRNA sequencing and suggest future research with a higher-resolution sequencing method in the discussion (Line 567):

“Third, due to the high degree of compositional and functional diversity within a genus,⁸⁷ we have limited insights into the possible functions that these microbes may perform. Future studies using higher resolution sequencing methods (i.e., to the species and strain level) and examining other “-omics” layers (e.g., transcriptomics, proteomics) can provide additional information about the specific functional processes within the early childhood gut microbiota that are related to brain development and internalizing symptoms in middle childhood.”

It would be helpful to test for differences in socio-demographics and variables of interest for participants that are included vs excluded from the sample.

We now report descriptive statistics and tests for differences in socio-demographics and variables of interest between the sample of included participants and those that were excluded based on the fMRI quality control in Table 1. The only significant difference found between the included and excluded participants is that excluded participants had higher average mean framewise displacement in their fMRI scan data, which is expected given that framewise displacement was one exclusion criterion in fMRI quality control (described in the methods). We now include a section in the SI describing these differences (starting line 201):

“Differences Between Included and Excluded Samples

T-tests for continuous variables and chi-square tests for categorical variables were used to evaluate bivariate differences in variables of interest and covariates for the analytic sample compared to the sample of children that participated in the year 6 fMRI scan, but whose data were excluded in fMRI quality control (Table 1). The only significant difference between included and excluded children was that excluded children had higher average mean FD, which is expected given that a mean FD cutoff was used as an exclusion criterion in the fMRI quality control (see “MRI Data Preprocessing”).”

In addition to including further investigation into the participants flagged as outliers.

We examined the two participants who were identified as outliers on brain networks, and the participant who was identified as an outlier on microbiome genera. Results of these analyses are now described in the SI (page 6, starting line 245) and reproduced here:

“The two participants identified as outliers on several rsFC network values were not outliers on any other continuous variables of interest or demographic characteristics. However, they did belong to some of the more uncommon demographic categories - one of the two outlier participants took antibiotics at age 2 years (13% of sample) and the other took probiotics at age 2 years (5% of sample).”

The participant identified as an outlier on abundance of several microbiome genera was also a high outlier on one of the alpha diversity metrics - observed features. Otherwise, this child was

not an outlier on any other continuous variables of interest or demographic characteristics, and was not a member of any uncommon demographic categories.”

Given the results of our outlier investigation, we also performed sensitivity analyses excluding children whose caregiver reported they had taken antibiotics and/or probiotics at the time point concurrent to the stool sample (reported in the SI and summarized here). These analyses involved repeating the multiple linear regression analyses described in the main text, in the subsample without antibiotic or probiotic exposure (N=47 out of 55). Those analyses found that associations between Brain Signatures and caregiver-reported internalizing symptoms, Microbial Profiles and Brain Signatures, and Microbial Profiles and caregiver-reported internalizing symptoms were virtually unchanged from main analysis results except that Microbial Profile 2 was significantly associated with caregiver-reported internalizing symptoms controlling for covariates in the subsample, whereas there was no significant association in the full analytic sample.

Minor comments – Figure S1. Add labels to the other side of the correlation matrix

Thank you for the suggestion. We have edited Figure S1 so that labels are now on the top as well as the left side of the correlation plot.

Figure S2 and Table S3 can be combined.

Thank you for the suggestion. We have added the information from Figure S2 to Table S3, combining the two.

Some of the network acronyms may not have been previously defined.

We appreciate the reviewer’s attention to detail. We have gone through the manuscript to ensure that all network acronyms have been defined before subsequent usage.

Reviewer #1 (Remarks on code availability): The code provides a nicely formatted R script. It would be helpful to have additional data describing the format of each of the data frames or even a blank dataframe with variable labels to further understand the structure of inputs.

In addition to the R script available at the provided GitHub link, we now also include at that link a data frame with simulated (i.e., fake) values for all variables (“data_simulated.csv”) so that the structure of inputs can be better understood.

Reviewer #2

This is an interesting study examining links between the child gut microbiome, brain networks, and internalizing symptoms later in childhood. There are several areas that could use some clarification or elaboration:

We thank the reviewer for these positive comments on the manuscript. We have addressed their comments below.

The manuscript may overstate the novelty of the tested associations. There is a robust body of research testing associations between microbiome, brain, and behavior early in development (including resting-state functional connectivity). The novelty is more evident in the longitudinal nature of their data and use of functional networks for brain outcomes.

Thank you for giving us the opportunity to better highlight the novelty of our research. The reviewer is correct that there are bodies of research testing associations between microbiome and behavior in development (van de Wouw et al., 2022, 2024), and among microbiome, brain (structural and functional), and behavior during infancy and toddlerhood (Carlson et al., 2021; Gao et al., 2019; Kelsey et al., 2021). Indeed, as the reviewer points out, the novelty in our study lies in the examination of prospective associations between the microbiome at 2 years of age, with resting state networks, and behavior beyond the infant years. Furthermore, we are unique in taking a symptom-centered approach to identify the specific functional architecture and microbes that best explain variability in childhood internalizing symptoms, providing potential targets for intervention. We elaborate on the significant gaps in the literature that the current study fills and the significant novel contributions of the current study in the manuscript, and essentially rewrote the entire introduction to include this information. As a result, we won't reproduce the whole introduction here, but will provide some notable excerpts:

Starting Line 119: *“Recent evidence that microbiome composition is associated with neurocognitive and behavioral outcomes during childhood, such as internalizing symptoms,^{13–15} underscores the importance of understanding the MGBA’s contribution to mental health during development. To date, however, most studies examining the MGBA in the context of typical development have focused on infancy and toddlerhood, with an emphasis on neurocognitive, rather than mental health, outcomes.^{16–19} These studies have found that individual differences in microbial alpha diversity are associated with variation in the structure and function of brain regions and networks implicated in sensorimotor processing, language, and emotion.^{16–19} For example, greater alpha diversity in infancy has been concurrently linked to increased fronto-parietal connectivity—a network implicated in cognitive control, which in turn was associated with increased infant negative emotionality.¹⁹ Similarly, higher alpha diversity at age 1 year has been associated with lower functional connectivity between the left amygdala and thalamus, and between the anterior cingulate cortex and insula, as well as greater connectivity between the supplemental motor area and inferior parietal lobule.¹⁸ Higher alpha diversity at age 1 year has also been linked with poorer cognitive performance (visual reception and expression language) and brain volume at later developmental stages—greater left precentral, right angular, and left amygdala volumes at age 2 years¹⁶—highlighting potential programming effects of early microbiota on later brain structure.*

In studies examining specific microbial taxa, concurrent and prospective associations with neurodevelopment have also been reported. For instance, Carlson et al.¹⁷ found that relative abundances of Streptococcus and Staphylococcus at age 1 month were concurrently negatively associated with amygdala volumes, and positively associated with medial prefrontal cortex (mPFC) volumes. In terms of potential programming effects, higher abundance of Bacteroides at age 1 month was linked to smaller mPFC at age 1 year, whereas greater

Lachnospiraceae abundance was prospectively associated with larger amygdala volume - an association that was reversed for *Enterobacteriaceae*.¹⁷ Notably, in that study, there were no concurrent associations between specific bacterial taxa and brain volumes at age 1 year, suggesting cascading developmental effects, whereby earlier microbial composition influences later brain structure. In that study, concurrent and prospective associations between microbes and affective outcomes were also reported: higher abundances of *Dialister*, and bacteria that are part of the Clostridiales order, at 1 year were associated with reduced fear behavior at 1 year, and alpha diversity at 1 month was associated with increased fear behavior at 1 year. These findings align with the hypothesis, also supported by research in rodents,²⁰ that early-life microbiota provide critical signaling inputs necessary for typical neurodevelopment, and that disruptions in these signals may confer risk for later psychopathology.⁶ Despite these insights, it remains unclear whether—and how—early life microbial diversity and composition affect brain function later in development, during middle childhood, a high risk period for mental health problems.^{4,21}”

Starting Line 205: “In the current study, we investigate how gastrointestinal microbiota composition at 2 years of age relates to children’s intrinsic functional brain network connectivity at 6 years of age, with these brain networks defined by their association with caregiver-reported internalizing problems at 7.5 years of age. We focus on internalizing problems for two reasons: First, internalizing symptoms during early-middle childhood increases risk for chronic and recurrent internalizing symptoms across the lifecourse,^{45,46} making this a particularly important symptom group to investigate for links to the microbiota. Second, recent evidence suggests that the associations between microbiota composition (alpha diversity and specific microbes) are more robustly associated with internalizing symptoms and somatic complaints than externalizing symptoms during the preschool years.¹⁴ This multivariate symptom-centered approach to identify specific brain functional architecture relevant to mental health, before linking it to the microbiome is a novel approach that can provide potential targets for future interventions. Furthermore, our examination of prospective links between early microbiota and brain networks/behavioral outcomes in middle childhood provides insight into the potentially long reach of early microbial composition on child outcomes, especially during periods of risk for the emergence of internalizing symptoms.^{47,48}”

Early childhood is a vast period with many different developmental periods that play a crucial role in how experiences are processed and how they affect the developing body and brain. It would help to add more details about the developmental periods in the review of existing literature. For example, on p. 4 there are references to several studies “... microbiome composition is associated with neurocognitive and behavioral outcomes during childhood, such as internalizing symptoms”; “... microbiome composition and differential abundance of single microbes, in infancy and toddlerhood” and “studies of children show that the functional neural connectome..”. Stating the specific age ranges of the children in these studies, as well as whether they were cross-sectional or longitudinal, would provide necessary information to determine if the current investigation (and the developmental periods chosen) is an appropriate and novel next step.

We appreciate the reviewer’s suggestion to include more details about what is currently known about the interplay between the microbiota, brain, and behavior across developmental periods.

We now include more details on prior research, including the specific ages of children, to contextualize the novelty of the current investigation (see our excerpts from the introduction in response to previous comment).

Better justification is needed for why resting-state functional data is relevant for clinical outcomes and the gut microbiome.

We now include justification in the introduction for why resting state functional data is relevant for clinical outcomes and gut microbiota in children:

Starting Line 157: *“Of the developmental studies linking the microbiota to brain function, most have employed seed-based resting state functional connectivity (RSFC) analyses.^{16,18} While use of RSFC is preferable in developing populations due to reduced task demands, the use of researcher-selected seeds in these analyses provide a limited view of brain function, and further, does not account for connectivity within and between canonical large-scale functional brain networks, which have been linked to neurocognitive behaviors and psychopathological outcomes in children.^{22–28} For example, internalizing symptoms in children have been associated with alterations in RSFC within the default mode network (DMN), and the ventral attention network (VAN), and between both DMN and striatal regions with cingulo-opercular (CON) networks.²² Across development, a pattern of increasing within-network integration and between-network segregation supports the increasing specialization of network-related functions and efficient information processing—deviations from which are associated with psychopathology in youth.^{29–33} As such, it is clear that investigating the impact of early microbiota on the development of brain network integration and segregation is an essential step for understanding the microbiota's contributions to psychopathology. However, such analyses are challenging due to the multivariate complexity of both microbiota and large-scale brain network data. Use of multivariate machine learning analysis techniques may help to solve this issue.”*

Neuroimaging at this age is difficult, and only 55 of 110 had usable imaging data. More details about missing data would help to evaluate whether this missingness was random or related to primary variables – ie., were there differences in child characteristics between those with usable data vs not? For example, are internalizing children more compliant and follow directions better than externalizing (or vice versa, internalizing children may be more nervous about being in the scanner). This may lead to bias with internalizing children being over or underrepresented in the sample.

We now report descriptive statistics and tests for differences in socio-demographics and variables of interest between the sample of included participants and those that were excluded in fMRI quality control in Table 1. The only significant difference found between the included and excluded participants is that excluded participants had higher mean framewise displacement in their fMRI scan on average, which is expected given that framewise displacement was one exclusion criterion used in the fMRI quality control.

It would be helpful to assess the model fit of the mediation analyses. Since the authors are testing multiple gut microbiome metrics as predictors of the pathway, evaluating model fit could provide insight into how well the mediation model captures the overall data patterns.

Given the complexity of the associations between the gut microbiome and the brain, poor model fit—regardless of statistical significance—could indicate issues with the magnitude of the observed effects.

Thank you for voicing the importance of examining mediation model fit to the data. We aren't able to compute typical measures of model fit (e.g., RMSEA, CFI) because simple mediation models with observable data (i.e., without any latent variables and all pathways specified) are fully saturated (i.e. degrees of freedom equal zero). As such, the model will fit the data perfectly by definition, and fit indices are not informative (Hayes, 2022; Kline, 2016; MacCallum & Austin, 2000). However, to emphasize magnitude of effects, we now report partial r-squared for the predictor of interest in each mediation model path (shown in Table S4), as a way to indicate how much variation in the outcome variable is explained by the predictor of interest in each model.

We also compared those values to other studies to assess whether the amount of variance explained is reasonable (now included in the SI and reproduced below):

SI page 9, starting line 367: *“We compared partial R² values from each path in the mediation models (Table S5) to other studies to assess whether the amount of variance explained in each path is reasonable. Partial R² for the a paths (gut microbiome to brain) in our study ranged from 0.013 to 0.29, which is similar to values in other studies on the gut microbiome and neuroimaging in childhood that reported partial R².^{8,29} Partial R² for the b paths (brain to internalizing symptoms) ranged from 0.06-0.22, which is similar to or higher magnitude than the other study that reported on rsFC and internalizing symptom associations in childhood.³⁰ The higher values we found could be because Albertina et al. examined individual pairs of ROIs within- and between-networks, whereas our analysis approach (i.e., generating brain signatures using sPLS) incorporates information from multiple ROI pairs in the same model. Based on this comparison, we believe that our models are adequately capturing the patterns in the data.”*

If there is another specific approach that you had in mind for assessing model fit of the mediation models, please let us know and we would be happy to implement it in the next round of revisions.

Although it's hard to avoid, the use of CBCL as the only outcome variable is not ideal. Parent-report of child behavior comes with clear limitations (subjective perceptions, bias, embarrassment) and does not reflect clinical significance. This should be included as a limitation – and perhaps consider using the phrase “parent reported internalizing symptoms” throughout to be clear that this is not a clinical diagnosis.

We have changed the wording to “caregiver-reported internalizing symptoms” throughout the manuscript to make it clear that the symptom outcome measure is parent-reported.

While we only had access to caregiver-reported internalizing symptoms at 7.5 years of age, the children in the study self-reported their depressive and anxiety symptoms using the Child Depression Inventory (CDI-2) and MASC, respectively, when they were approximately 8.5 years of age. We conducted supplemental analyses (reported in the SI) relating the brain and microbial

components to children's self-reported depressive and anxiety symptoms at 8.5 years and found that the brain components also related to children's self-reported depressive symptoms, demonstrating that our findings are robust to reporters.

Moreover, in this age group, clinician assessment tends to agree more with parent-report than child-report (Hawley & Weisz, 2003), indicating that parent report can indeed reflect clinical significance of symptoms, and is the primary reason that we used parent-reported child symptoms as our primary outcome measure in this paper. We have included more justification for our choices in the methods:

Starting Line 608: *"We used parent-reported child symptoms at 7.5 years on the Child Behavior Checklist (CBCL)89 as our primary indicator of child mental health. Parent-report was chosen over child self-report as clinician assessment tends to agree more with parent-report than child-report in the age range of this study."⁴⁹*

A major concern with the covariate selection is that the authors primarily chose covariates based on prior fMRI and microbiome literature. However, for the diet-related covariates, they used a data-driven approach, which seems contradictory. For example, several potential covariates—such as maternal age, breastfeeding, or formula feeding—could have yielded significant results outside of the diet analysis if tested. The covariate selection process should be consistent throughout.

We have reworded our description of covariate selection, as it was confusing and may have led to a misunderstanding. Actually, we chose to control for diet *a priori* based on prior microbiome literature showing dietary influences on childhood gut microbiome composition (Herman et al., 2020). However, in our study we had 8 indicators of diet. Due to our small sample size, we used a data-driven approach to narrow down the number of diet indicators to include as covariates in order to preserve power. We've changed the methods section to clarify our approach. It now reads as follows (page 16-17):

Starting line 665: *"To be consistent with prior fMRI and gut microbiota psychopathology research in the GUSTO study cohort,^{44,96} covariates were selected for inclusion a priori. These included sex, gestational age, birthweight, mean framewise displacement (MRI), delivery mode, maternal education, and diet. While 3 indicators of diet were ultimately used as covariates - consumption of fiber, total fat, and polyunsaturated fat - there were in fact a total of 8 diet indicators from which the final 3 indicators were selected: consumption of total fat, polyunsaturated fat, monounsaturated fat, saturated fat, fiber, carbohydrates, and protein in the diet, and age that the child stopped breastfeeding. To select covariates from the pool of 8 diet indicators, we included those indicators that were significantly related to the microbiota, brain, and/or caregiver-reported internalizing symptoms. See SI for more details on diet covariate selection process and significant associations."*

Authors should do multiple comparison correction in their regression analyses.

We have changed our analyses on associations between gut microbiota alpha diversity and the derived brain components to control for multiple comparisons, since we examined 4 indicators of

alpha diversity in that set of analyses. The description of alpha diversity analyses in the methods (page 18, starting line 736) now includes the following note on multiple comparison correction:

“Because we had 4 indicators of alpha diversity, statistically significant associations were subjected to multiple comparison correction using the Benjamin-Hochberg method,¹⁰¹ with a q-value threshold for significance of 0.25, as is recommended for biomarker discovery investigations.^{19”}

We do not correct for multiple comparisons in the other multiple regression analyses, which are associations between derived brain signatures and internalizing symptoms and derived microbial profiles and brain signatures, because those regression analyses were robustness checks to confirm whether the components derived in sPLS were still significantly associated with the “Y” variable used in the sPLS while controlling for covariates, since we could not include covariates in the sPLS analysis directly.

Reviewer #4

The manuscript "Childhood gut microbiome is linked to mental health at school age via the functional connectome" describes links between gut microbiota at 2 years, connectome at 6 years, and internalizing symptoms at 7.5 years. While combining multiple data types is very interesting and valuable, I am afraid that the models are overfitted due to the low sample size. Could you test with some other method(s) if the results and interpretations are robust?

Thank you for your kind comments about our manuscript, and for raising the concern about overfitting the models. As described in the methods, we validated each sPLS model using cross-validation, meaning that we selected the solution that maximized the average R^2 across cross-validation folds and repeats (we used 10-fold cross-validation repeated 100 times) to achieve a solution that was robust to perturbations of the sample. For each feature selected in our sPLS models, we now also report stability values in Tables S2 (brain signatures) and S3 (microbial profiles). Stability values represent how frequently the feature was selected for inclusion across cross-validation repeats, ranging from 0 (none of the repeats) to 1 (100% of the repeats). In the results and discussion sections, we focus only on features with high stability (≥ 0.8), as is recommended (Le Cao & Welham, 2022). We’ve added text to the methods section describing our approach to stability:

Starting Line 700: *“sPLS regression models were tuned for the optimal number of brain network components and variables per component using 10-fold cross validation repeated 100 times. The model that maximized cross-validated R^2 was selected as best-fitting. In the context of sPLS, R^2 is defined as the correlation between observed caregiver-reported internalizing symptoms and internalizing symptoms predicted by the model.¹⁰⁰ Outliers (third/first quartile $\pm 1.5 \cdot IQR$) on any network were winsorized for that network, and model tuning was repeated on the winsorized dataset. Outlier identification details, and other characteristics of participants flagged as outliers, can be found in the SI. For each final tuned model, we report a Pearson’s correlation between scores on the derived signature from the final model with the outcome variable, the*

loading of each network onto each component (which measures the relative strength of each network's contribution and the direction of its association with internalizing symptoms), the Variable Importance in the Projection (VIP) of each network onto each component (which represents the percent of variance in caregiver-reported internalizing symptoms that is explained by the network divided by the total variance explained by the component), and the stability of each network (which represents how frequently the network was selected for inclusion across cross-validation repeats, or the reproducibility of the derived "signatures" when the training set is perturbed). We focus our interpretation on networks with stability 0.8, as is recommended.¹⁰⁰"

We have now also added supplemental analyses (described in the results section) examining whether the derived brain signature and microbial profiles, in addition to being related to caregiver-reported internalizing symptoms, would also be associated with other measures of child mental health, including child self-reported mental health at a later time point (8.5 years of age) and caregiver-reported externalizing symptoms (as you can see in response to Reviewer #1).

Moreover, better inferences on the functional potential or output of the gut microbiome would be more informative. The authors do acknowledge this in the discussion, and if gaining new data - such as faecal metagenome or serum metabolomics or so forth - is not feasible, I would recommend predicting functional pathways based on the amplicon sequencing data. It often helps with the inference, and you could even compute sPLS similarly of the predicted metagenome.

Thank you for the suggestion. In this revision, we predicted functional pathways from our 16S sequencing data with PiCRUST 2, and performed similar sPLS analyses to identify predicted functional microbial profiles that covaried most strongly with our derived brain signatures.

Now reported in the SI, we found one functional profile covaried most strongly with the SOFA, MTL, SAL, PMN Network Connectivity Brain Signature, and remained significantly associated with its corresponding brain signature controlling for covariates. The pathway that had high stability from this component, which also had the strongest loading, was PWY-6270, isoprene biosynthesis I (negative loading). We also found two functional profiles covaried most strongly with the SOFA Between Network Connectivity Brain Signature, one of which remained significantly associated with its corresponding brain signature while controlling for covariates. Pathways from these profiles with the highest loadings, and high stability, included the pentose phosphate pathway and superpathway of glucose and xylose degradation (negative loadings), and cob(II)yrinate a,c-diamide biosynthesis I pathway, NAD salvage pathways V and I, and UPD-N-acetyl-D-glucosamine biosynthesis I pathway (positive loadings).

As we now explain in the discussion, this analysis suggests that microbial functions involved in cellular energy metabolism may contribute to brain development that is relevant for internalizing symptoms at school age.

Starting Line 503: *"While our taxonomic analyses suggested that microbes involved with inflammation, physical, and mental health symptoms were explaining most variance in the internalizing symptom associated brain signatures, supplemental analyses that identified profiles*

of predicted microbiome functional pathways most strongly linked to the derived brain signatures found the strongest associations with pathways involved in cellular energy metabolism (nicotinamide adenine dinucleotide salvage and phosphate pathways).^{84,85} Future research using functional potential derived from shotgun metagenomics is needed to verify these findings given the established limitations of inferring functional potential from amplicon sequencing data.⁸⁶ ”

Thirdly, the authors would have access to a wider array of psychiatric symptom domains, and I would believe that there would be many relevant outcomes related to the gut-brain axis. Could the authors provide better rationale on why you particularly focused on internalizing symptoms and not e.g. inattention?

This comment was similar to one made by Reviewer #1. As stated in the response to that reviewer: While we acknowledge that the MGBA has been implicated in a range of mental health outcomes, the scope of the current investigation focused on identifying specific microbiota and functional network-level predictors of childhood internalizing symptoms. The emergence of internalizing symptoms during early-middle childhood not only increases the risk for chronic and recurrent internalizing symptoms across the lifecourse, but recent evidence suggests that the associations between microbiota composition (alpha diversity and specific microbes) are more robustly associated with internalizing symptoms and somatic complaints than externalizing symptoms during the preschool years (van de Wouw et al., 2022).

In supplemental analyses, we tested whether the microbiota and brain features that we identified to be predictive of internalizing symptoms were driven by anxiety or depressive symptoms and/or also predictive of, and thus generalizable to, externalizing symptoms. We found that the SOFA, MTL, SAL, PMN Network Connectivity Brain Signature was positively associated with caregiver-reported externalizing, depression, and anxiety symptoms, as well as with child self-reported depressive symptoms at a later time point, whereas the SOFA Between Network Connectivity Brain Signature was only associated with child self-reported depressive symptoms at a later time point. Similarly, Microbial Profile 1 was positively associated with caregiver-reported externalizing symptoms; Microbial Profiles 2 and 3 were not associated with any additional symptom measures (results reported in the SI). Though most of these associations were not robust to the inclusion of covariates, they suggest that the SOFA, MTL, SAL, PMN Network Connectivity Brain Signature and Microbial Profile 1 may capture contributions that are related to psychopathology generally, whereas the SOFA Between Network Connectivity Brain Signature and Microbial Profiles 2 and 3 may capture contributions that are more specific to internalizing symptoms. Future studies should use a similar symptoms-based approach for other domains of functioning to identify symptom-specific microbiota-brain profiles. We include these analyses in the revised SI (pages 6-7, starting line 255):

“Brain Signature Associations with Other Mental Health Domains and Reporters

Scores on the SOFA, MTL, SAL, PMN Network Connectivity Brain Signature were positively associated with caregiver-reported externalizing symptoms ($r=.39$, $t(53) = 3.06$, $p=.003$), depressive symptoms ($r=.36$, $t(53) = 2.81$, $p<.001$), and anxiety symptoms ($r=.38$, $t(53) = 2.98$, $p=.005$). They were also positively associated with child self-reported depressive symptoms ($r=.47$, $t(30) = 2.91$, $p=.005$), but not anxiety symptoms ($r=-.22$, $t(31) = -1.23$,

$p=.18$), at age 8.5 years. The association with caregiver-reported anxiety symptoms was robust to inclusion of covariates ($\beta = 0.42$, $SE = 0.19$, $p = .031$), but the other associations were not: caregiver-reported externalizing symptoms ($\beta = 0.44$, $SE = 0.23$, $p = .060$), depressive symptoms ($\beta = 0.45$, $SE = 0.28$, $p = .12$), and child self-reported depressive symptoms ($\beta = 0.75$, $SE = 0.36$, $p = .061$).

Scores on the SOFA Between Network Connectivity Brain Signature were not associated with caregiver-reported externalizing symptoms ($r=.29$, $t(53) = 2.21$, $p=.096$), depressive symptoms ($r=.10$, $t(53) = 0.74$, $p=.47$), or anxiety symptoms ($r=.22$, $t(53) = 1.67$, $p=.20$). However, they were positively associated with child self-reported depressive symptoms ($r=.32$, $t(30) = 1.87$, $p=.02$), but not anxiety symptoms ($r=.08$, $t(31) = 0.43$, $p=.56$), at age 8.5 years. The association with child self-reported depressive symptoms was not robust to the inclusion of covariates ($\beta = 0.31$, $SE = 0.51$, $p = .56$).

Microbial Profile Associations with Other Mental Health Domains and Reporters

Scores on Microbial Profile 1 were positively associated with caregiver-reported externalizing symptoms ($r = .20$, $t(53) = 1.71$, $p = .033$), but not anxiety symptoms ($r = .06$, $t(53) = 0.47$, $p = .66$) or depressive ($r = .09$, $t(53) = 0.64$, $p = .44$) symptoms. They were also not associated with self-reported depressive ($r = -.22$, $t(30) = -1.25$, $p = .29$) or anxiety ($r = .02$, $t(31) = 0.091$, $p = .94$) symptoms at a later time point (age 8.5 years). The positive association with caregiver-reported externalizing symptoms was not robust to inclusion of covariates ($\beta = 0.20$, $SE = 0.18$, $p = .26$).

Scores on Microbial Profile 2 were not associated with caregiver-reported externalizing symptoms ($r = .10$, $t(53) = 0.76$, $p = .51$), anxiety symptoms ($r = .14$, $t(53) = 1.05$, $p = .35$), or depressive symptoms ($r = .05$, $t(53) = 0.39$, $p = .65$); or self-reported depressive symptoms ($r = .04$, $t(30) = 0.21$, $p = .84$) at a later time point (age 8.5 years). Scores were marginally positively associated with self-reported anxiety symptoms ($r = .29$, $t(31) = 1.70$, $p = .080$) at a later time point (age 8.5 years).

Scores on Microbial Profile 3 were marginally positively associated with caregiver-reported externalizing symptoms ($r = .21$, $t(53) = 1.58$, $p = .090$). Scores were not associated with caregiver-reported anxiety symptoms ($r = .11$, $t(53) = 0.78$, $p = .45$) or depressive symptoms ($r = .11$, $t(53) = 0.80$, $p = .33$); or self-reported depressive symptoms ($r = .14$, $t(30) = 0.77$, $p = .45$), or self-reported anxiety symptoms ($r = -.37$, $t(31) = 1.51$, $p = .18$) at a later time point (age 8.5 years)."

And mention them in the manuscript discussion section (page 13, starting line 533):

"However, our supplemental specificity analyses suggested that the brain signatures and microbial profiles we found in this study were not all linked specifically to internalizing symptoms in youth. The SOFA, MTL, SAL, PMN Network Connectivity Brain Signature and Microbial Profile 1 seemed to capture contributions that were related to psychopathology generally, whereas the SOFA Between Network Connectivity Brain Signature and Microbial Profiles 2 and 3 seemed to capture contributions that were more specific to internalizing symptoms. As such, the symptom anchored approach we used here had the intended benefit of identifying biological features that were relevant to our outcome of interest - internalizing symptoms, as well as in some cases to broader mental health outcomes. Beyond approaches that combine multidimensional biological datasets without reference to symptoms, incorporating a

symptom anchored approach can accelerate biomarker discovery by identifying symptom-relevant brain and microbial features from the outset, as opposed to identifying the most highly correlated/covarying features in general, which may or may not be symptom relevant.”

To add to the clinical characterization of the sample, we also now include descriptive statistics on caregiver-reported externalizing, depression, and anxiety symptoms in Table 1, as well as the percentage of the sample exhibiting clinically significant levels of caregiver-reported internalizing and externalizing symptoms based on CBCL t-score clinical cutoffs.

Minor comments:

Since the manuscript contains amplicon sequencing data, the correct term would be microbiota, not microbiome.

We have now changed all instances of microbiome to microbiota in the manuscript.

The introduction seems to be a little cumbersome to read, and I wonder whether you could focus a little less on the different modeling options, since this is not a methodological paper anyway?

Given that one of the important innovations in our study is the use of advanced multivariate approaches that integrate the complex multivariate structures of both the microbiota and brain networks, a fact recognized by several other reviewers, we have elected to retain information on modeling in the introduction. However, we have rewritten the introduction to streamline it and increase understanding.

I would repeat the sample size (with and without covariates) and description of the sample in the beginning of the results. I would like to read on the variation of symptoms, inclusion and exclusion criteria, and information on key confounders, such as antibiotic and other medication use, psychiatric diagnoses, and if there were any high-risk families (SES, income, parental psychiatric diagnoses) and so forth.

We have added a description of inclusion and exclusion criteria to the SI (in section “Participants and Study Design”). We have also added a short description of the sample to the start of the results (new section, “Descriptive Statistics”, page 7, starting line 248):

“The analytic sample for this study included N=55 children who contributed a usable stool sample at age 2 years, good-quality fMRI data at age 6 years, and whose caregiver completed an assessment of their caregiver-reported internalizing problems at age 7.5 years. Of those, N=43 who provided data on all covariates were included in analyses that controlled for covariates. Descriptive statistics for the analytic sample and the sample of children excluded due to poor quality fMRI data (N=55) are shown in Table 1. The only difference between the included and excluded samples were that excluded participants had higher mean framewise displacement on average, which was the reason for their exclusion (Table 1). “

We have added information about family SES (maternal education), household income per member, maternal depression and anxiety symptoms, antibiotic and probiotic use, other

medication use, and child mental health (caregiver-reported internalizing, depressive, anxiety, and externalizing symptoms raw scores and percent of sample above clinical cutoff) to Table 1 and to the supplement. Relevant information is summarized below:

While data on child psychiatric diagnoses was not collected, we report the percentage of children at age 7 years showing elevated levels (borderline or clinical) of:

- Caregiver-reported internalizing symptoms (16%)
- Caregiver-reported externalizing symptom (10%)

Table 1 also provides information on high-risk families:

- 9% of mothers had moderate or severe levels of depressive symptoms at child age 6 years
- 18% of mothers had clinically elevated levels of anxiety symptoms at child age 6 years
- 62% of mothers reported their highest level of education was less than a university degree. However, 95% of those who responded reported at least a secondary education, which is similar to population-level rates for reproductive-age adults (ages 25-34) in the 2010 Singapore census (93.9%, (*A New Educational Perspective: The Case of Singapore / Penn GSE Perspectives on Urban Education*, n.d.)
- 24% reported family income below a commonly used cutoff for the poverty line in Singapore (Donaldson et al., 2013)

And exposure to antibiotics and probiotics:

- 13% of the sample were reported to be taking antibiotics at age 2 years (timepoint concurrent to microbiome assessment)
- 5% of the sample were reported to be taking probiotics at age 2 years

Because a small subset of the sample reported exposure to antibiotics or probiotics, we performed sensitivity analyses excluding children whose caregiver reported they had taken antibiotics and/or probiotics (reported in SI and summarized here). These analyses involved repeating the multiple linear regression analyses described in the main text, in the subsample without antibiotic or probiotic exposure (N=47 out of 55). Those analyses found that associations between Brain Signatures and caregiver-reported internalizing symptoms, Microbial Profiles and Brain Signatures, and Microbial Profiles and caregiver-reported internalizing symptoms were virtually unchanged from main analysis results except that Microbial Profile 2 was significantly associated with caregiver-reported internalizing symptoms controlling for covariates in the subsample, whereas there was no significant association in the full analytic sample.

In addition, since the results are before methods, please help the reader and list the covariates used in the analyses.

We now include a description of covariates used in the analyses in the introduction (page 7, lines 242-244):

“For all analyses, we covaried for child sex assigned at birth, gestational age, birthweight, mean framewise displacement (motion in fMRI), delivery mode, maternal education, and diet.”

It is also helpful for a non-expert reader to open the abbreviations when first mentioned.

We have checked the manuscript to make sure to define the abbreviations when they are first mentioned.

Since there is no repeated assessment of microbiome, psychiatric symptoms or connectome, I believe the correct term is a cross-sectional study (in a longitudinal cohort study though).

We respectfully disagree. The core characteristic of a longitudinal study is that the same people are followed over time. Repeat measurements are not an absolute requirement of a longitudinal design. This is a prospective longitudinal study.

Can you explain in brief the gut microbiota methodology in the main text as well? In addition it is great to know if you estimated ASVs, the taxonomy data based and so forth. I would also report for instance in the supplements if you used any controls.

We have added information on controls to the supplement, and have brought the other information on microbiota methodology (whether we estimated ASVs, taxonomy database used, etc.) into the main text from the supplement.

Starting Line 632: *“Gut microbiota collection, storage, DNA extraction and sequencing steps are described in the SI. Reads were clustered into operational taxonomic units (OTUs) using USEARCH v9.2.64 at a 97% similarity threshold, and taxonomic classification was assigned to each OTU by comparing against the SILVA 123 ribosomal reference database.”*

Starting Line 643: *“As is expected for 16S rRNA sequencing, most sequences were classified to the genus level. OTUs were aggregated to the genus level for downstream analysis. Where a genus-level classification was not available, the family-level classification was used instead.”*

In the supplements, I cannot really parse the sentence "We used alpha diversity as this microbiome feature was not derived from the brain or internalizing symptoms."

Thank you for pointing out that confusing sentence. We have now re-worded it:

Page 4 (lines 172-174): *“We used alpha diversity as the microbiota variables for diet covariate selection because this microbiome feature was not derived from the brain or internalizing symptoms (in contrast to the microbial profile scores).”*

I commend the authors for sharing the analytical pipeline. I would also recommend using the STORMS checklist.

We have completed the STORMS checklist and added it to the study’s materials as another supplemental document, and have added information from the STORMS checklist that was missing from the original version of the revised manuscript to the manuscript and SI. Added information is noted in the STORMS checklist supplemental document with comments, and reproduced below:

Methods section, page 16-17, starting line 635: “Microbiota data were normalized using rarefaction (depth cutoff = 5,777) to account for uneven sequencing depth between samples (ten samples were removed due to having a sequencing depth shallower than cutoff)...”

SI, page 1, lines 17-18: “Data relevant to this paper was collected in 2011-2013 (stool samples), 2015-2017 (fMRI scans), and 2017-2018 (caregiver-reported internalizing symptoms).”

SI, page 3, lines 126-128: “No controls or replicates were used. No methods were used to control for or identify contamination, but all samples had sufficient microbial load, making contamination less relevant. A taxonomic classification was assigned to each OTU by comparing against the SILVA 123 ribosomal reference database (<https://www.arb-silva.de/>). There were no differences in any study variables of interest across the two sequencing batches.”

SI, page 3, line 114-115: “No preservatives or buffers were added for storage.”

Reviewer #5

The study investigates the influence of infant microbiome composition on later brain and behavioral development using a sequential sparse Partial Least Squares (sPLS) approach. The authors identify key microbial signatures associated with internalizing symptoms, mediated through brain networks. I have several concerns regarding its design and methodology. In the introduction, the authors should provide additional context regarding the stability and development of the microbiome from ages 2 to 8, as well as the developmental trajectory of the connectivity measures used. This would help readers better understand the rationale behind examining microbiome composition at 2 years in relation to resting-state functional connectivity (RSFC) and behavioral outcomes measured at 6–7.5 years.

We thank the reviewer for this suggestion. We now include a brief description of the stability and development of the gut microbiota during childhood and the development trajectories of the resting state connectivity networks. We also elaborate on the rationale behind examining microbiota composition at 2 years in relation to RSFC and behavioral outcomes at 6-7.5 years, citing experimental studies demonstrating the role of the early gut microbiota in shaping normal brain development:

Page 4, starting line 111: “Early childhood is a rapid period of gastrointestinal microbiome maturation.^{5,6} The microbiome reaches an adult-like configuration by around 3 years of age,^{5,7,8} after which point it remains relatively stable.⁹⁻¹¹ This early period may represent a sensitive window during which microbial signals shape key aspects of brain development.^{6,12} Such early influences could have lasting effects by programming neurobiological systems in ways that confer vulnerability or resilience to later mental health outcomes, particularly during the high-risk period of middle childhood.^{2,3} A developmental perspective on the MGBA is therefore essential for understanding how early microbial signals may contribute to lifelong trajectories of mental health.”

Page 5, starting line 166: “Across development, a pattern of increasing within-network integration and between-network segregation supports the increasing specialization of network-related functions and efficient information processing—deviations from which are associated with psychopathology in youth.^{29–33} As such, it is clear that investigating the impact of early microbiota on the development of brain network integration and segregation is an essential step for understanding the microbiota's contributions to psychopathology. However, such analyses are challenging due to the multivariate complexity of both microbiota and large-scale brain network data. Use of multivariate machine learning analysis techniques may help to solve this issue.”

In the covariates, the authors do not mention any SES variables. Why was that not included given its role in affecting all three domains the authors tested.

Thank you for this suggestion. We elected to use maternal education, considered to be one of the most influential indicators of SES (Khalatbari-Soltani et al., 2022; Lindberg et al., 2022), as a covariate. We elected to not use household income as a covariate due to a very strong positive relationship between maternal education level (less than a university degree vs. a BA degree or higher) and income [$t(34.9) = 4.28, p < .001$, Cohen's $d = 1.23$], but we now report on both income and maternal education in Table 1.

The use of a sequential sPLS approach with a relatively small sample size (n = 55) raises concerns about the robustness of the findings. Was stability selection implemented to ensure that the selected features are consistent across bootstrapped models? Without such measures, there is a significant risk of overfitting as mentioned in their limitations.

We were aware of the stability selection option, but we purposely chose not to perform stability selection since it biases the model. Specifically, if we tune the model based on feature stability, and then compute the final model parameters using only the stable features, this results in a selection bias (i.e., selecting only stable variables). The variables you select have already been 'seen' by the algorithm and testing the performance of the model with those specific variables can be overly optimistic, or wrong.

Ideally, if we had additional independent datasets, we could perform stability assessment of the features selected on the first training dataset and then select only those features to derive a final model in a second training set and then test in a 3rd data set. We do not have multiple test and training datasets to perform this iterative process, which is a limitation of our study:

Page 14, lines 558-560: “Nonetheless, there remains a small risk that the model was overfitted and our findings should be replicated in larger samples with separate training and testing sets to evaluate generalizability.”

However, to better address feature stability, in line with the reviewers comment, we have calculated, and now report, stability of all features selected in our models (reported in Table S2 for brain signatures and Table S3 for microbial profiles). You can see that the variables contributing the most to the models (i.e., ones with highest loadings) also have high stability; so,

the features with low stability are likely to be less influential on the sPLS component scores. Further, in our results and discussion sections, we focus interpretation exclusively on the features with high stability (≥ 0.8 out of 1.0, as has been recommended; (Le Cao & Welham, 2022; Meinshausen & Bühlmann, 2010).

Additionally, selecting the best-fitting model based solely on R^2 in a high-dimensional dataset with a limited sample size is concerning, as it may lead to R^2 inflation due to overfitting. I strongly recommend reporting cross-validated R^2 or adjusted R^2 to provide a more reliable measure of model performance.

We did, in fact, use cross-validated R^2 as our measure of model performance. We regret that this point was not clear in the original manuscript. We've now revised the methods section to make that apparent. It now reads:

Page 17, lines 700-702: “sPLS regression models were tuned for the optimal number of brain network components and variables per component using 10-fold cross validation repeated 100 times. The model that maximized cross-validated R^2 was selected as best-fitting.”

Furthermore, I have concerns of the use of a sequential sPLS modeling approach. Given the small sample size, errors or biases introduced in the first sPLS step (linking neuroimaging data to CBCL scores) are likely to propagate to the subsequent step. Would an integrative approach, such as DIABLO, which analyzes all datasets simultaneously, be more appropriate? I encourage the authors to explore this alternative and compare the results.

Thank you for this comment. We did consider using DIABLO as our analytical approach but elected not to for the following reasons. DIABLO is used to predict or classify categorical outcome variables, whereas our outcome variable (internalizing symptoms) is continuous. DIABLO is used with categorical variables because the method for selecting the optimal number of variables per input dataset with a continuous outcome variable is complex, and the developers have not yet released a tuning function for continuous outcomes (Le Cao, 2023). We avoided dichotomizing our internalizing symptom outcome variable because doing so would lose important variability captured by the continuous variable, and furthermore, it is unclear what an appropriate cutoff would be, as most participants in our sample did not have clinically elevated symptom levels.

In our paper, we identify the brain networks that are most strongly related to caregiver-reported internalizing symptoms, and then see which microbial genera are most strongly related to participant scores on internalizing symptom-associated brain network signatures. Particularly because our study design is prospective (i.e., including microbiome data from 2 years of age, brain data from 6 years of age, and internalizing symptoms from 7 years of age), a stepwise approach can provide insight into how the microbiome might influence subsequent brain development, and in turn how brain development might then influence internalizing symptoms.

To address the reviewer's concern about the propagation of biases, we have now added information on derived signatures' and microbial profiles' associations with other child mental health measures (see response to reviewer #1). Both brain signatures were significantly related to

self-reported depressive symptoms at age 8.5 years, suggesting that their associations with mental health are robust to reporter and the measure used in their derivation. Additionally, we now report on the stability of selected features on each brain signature and microbial profile, and find that the most stable features selected by each model (i.e., the ones that were selected for inclusion across cross-validation repeats) contribute the most to the signatures/profiles, whereas unstable features have low loadings (see response to reviewer #5's earlier comment about stability selection).

Regarding the selection of features for the second sPLS step, the authors report using the features with the highest loadings on the first two components. Was the latent score used as an input for the next sPLS step? If not, I would like to see those results included for comparison.

We regret that this important piece of information was not more obvious. Yes, the latent component scores were always used as input for the next sPLS step. We've now revised the methods section to make that more apparent. The relevant sections now read as follows, for brain signatures:

Page 17-18, lines 718-719: *“Following tuning, derived latent component scores along with selected covariates were entered into multiple linear regression models predicting caregiver-reported internalizing symptoms”*

And for the microbial profiles - Page 18, starting line 725: *“Next, we used sPLS regression, using the same methods as described in the previous analysis step, to identify linear combinations of genera abundances (i.e., ‘microbial profiles’) at age 2 years (predictor dataset) that maximally covaried with the latent component scores from the ‘brain signatures’ (outcome variable) identified in the previous analysis step. Models were tuned and outliers were screened using the same procedures described in the SI. After microbiota component scores were derived, multiple linear regression models including latent microbiota component scores and covariates predicting latent component scores from each brain signature were run to evaluate statistical significance and robustness to covariate inclusion, as described in the previous analysis step.”*

Additionally, in the discussion, the authors should elaborate on the relevance of their findings. Specifically, they need to clarify the rationale for how microbiome composition at 2 years of age could influence brain connectivity measures at 6 years. Providing a justification for the chosen time points would help with the interpretation of the results and contextualize their implications.

We thank the reviewer for the opportunity to provide more context for our study and to elaborate on the relevance of our findings. We now include in the introduction evidence for early life gut microbiota influences on brain development, and highlight the gaps in current knowledge on this topic that our study addresses:

Page 4, starting line 144: *“Notably, in that study, there were no concurrent associations between specific bacterial taxa and brain volumes at age 1 year, suggesting cascading developmental effects, whereby earlier microbial composition influences later brain structure. In that study, concurrent and prospective associations between microbes and affective outcomes were also*

reported: higher abundances of Dialister, and bacteria that are part of the Clostridiales order, at 1 year were associated with reduced fear behavior at 1 year, and alpha diversity at 1 month was associated with increased fear behavior at 1 year. These findings align with the hypothesis, also supported by research in rodents,²⁰ that early-life microbiota provide critical signaling inputs necessary for typical neurodevelopment, and that disruptions in these signals may confer risk for later psychopathology.⁶ Despite these insights, it remains unclear whether—and how—early life microbial diversity and composition affect brain function later in development, during middle childhood, a high risk period for mental health problems.^{4,21}”

Overall, while this study presents an interesting contribution to our knowledge on microbiome-brain-behavior interactions, addressing these methodological concerns would strengthen the validity and interpretability of the findings.

We thank the reviewer for their kind comments regarding our manuscript and we agree that our addressing of these methodological concerns has strengthened both the validity and interpretability of the findings.

References in Responses

A New Educational Perspective: The Case of Singapore | Penn GSE Perspectives on Urban

Education. (n.d.). Retrieved May 3, 2025, from

<https://urbanedjournal.gse.upenn.edu/volume-14-issue-1-fall-2017-15-years-urban-education-special-anniversary-edition-journal/new>

Achenbach, T. M., & Rescorla, L. A. (2001). *Manual for Child Behavior Checklist for Ages 6-*

18. APA PsycTests. <https://doi.org/10.1037/t47452-000>

Booth, C., Moreno-Agostino, D., & Fitzsimons, E. (2023). Parent-adolescent informant

discrepancy on the Strengths and Difficulties Questionnaire in the UK Millennium

Cohort Study. *Child and Adolescent Psychiatry and Mental Health*, *17*(1), 57.

<https://doi.org/10.1186/s13034-023-00605-y>

Burt, S. A. (2009). Rethinking environmental contributions to child and adolescent

psychopathology: A meta-analysis of shared environmental influences. *Psychological*

Bulletin, *135*(4), 608–637. <https://doi.org/10.1037/a0015702>

Cao, K.-A. L., Rossouw, D., Robert-Granié, C., & Besse, P. (2008). A Sparse PLS for Variable

Selection when Integrating Omics Data. *Statistical Applications in Genetics and*

Molecular Biology, *7*(1). <https://doi.org/10.2202/1544-6115.1390>

Carlson, A. L., Xia, K., Azcarate-Peril, M. A., Rosin, S. P., Fine, J. P., Mu, W., Zopp, J. B.,

Kimmel, M. C., Styner, M. A., Thompson, A. L., Propper, C. B., & Knickmeyer, R. C.

(2021). Infant gut microbiome composition is associated with non-social fear behavior in a pilot study. *Nature Communications* *2021 12:1*, *12*(1), 1–16.

<https://doi.org/10.1038/s41467-021-23281-y>

Chun, H., & Keleş, S. (2010). Sparse partial least squares regression for simultaneous dimension

reduction and variable selection. *Journal of the Royal Statistical Society. Series B*,

- Statistical Methodology*, 72(1), 3–25. <https://doi.org/10.1111/j.1467-9868.2009.00723.x>
- Donaldson, J. A., Loh, J., Mudaliar, S., Kadir, M. M., Wu, B., & Yeoh, L. K. (2013). Measuring Poverty in Singapore: Frameworks for Consideration. *Social Space*, 6, 58–66.
- Egger, H. L., & Angold, A. (2006). Common emotional and behavioral disorders in preschool children: Presentation, nosology, and epidemiology. *Journal of Child Psychology and Psychiatry*, 47(3–4), 313–337. <https://doi.org/10.1111/j.1469-7610.2006.01618.x>
- Gao, W., Salzwedel, A. P., Carlson, A. L., Xia, K., Azcarate-Peril, M. A., Styner, M. A., Thompson, A. L., Geng, X., Goldman, B. D., Gilmore, J. H., & Knickmeyer, R. C. (2019). Gut microbiome and brain functional connectivity in infants—a preliminary study focusing on the amygdala. *Psychopharmacology*, 236(5), 1641–1651. <https://doi.org/10.1007/s00213-018-5161-8>
- Hawley, K. M., & Weisz, J. R. (2003). Child, parent and therapist (dis)agreement on target problems in outpatient therapy: The therapist’s dilemma and its implications. *Journal of Consulting and Clinical Psychology*, 71(1), 62–70. <https://doi.org/10.1037/0022-006X.71.1.62>
- Hayes, A. F. (2022). *Introduction to mediation, moderation, and conditional process analysis: A regression-based approach* (pp. xvii, 507). Guilford Press.
- Herman, D. R., Rhoades, N., Mercado, J., Argueta, P., Lopez, U., & Flores, G. E. (2020). Dietary Habits of 2- to 9-Year-Old American Children Are Associated with Gut Microbiome Composition. *Journal of the Academy of Nutrition and Dietetics*, 120(4), 517–534. <https://doi.org/10.1016/j.jand.2019.07.024>
- Hwang, J. W., Egorova, N., Yang, X. Q., Zhang, W. Y., Chen, J., Yang, X. Y., Hu, L. J., Sun, S., Tu, Y., & Kong, J. (2015). Subthreshold depression is associated with impaired resting-

- state functional connectivity of the cognitive control network. *Translational Psychiatry*, 5(11), e683–e683. <https://doi.org/10.1038/tp.2015.174>
- Jami, E. S., Hammerschlag, A. R., Ip, H. F., Allegrini, A. G., Benyamin, B., Border, R., Diemer, E. W., Jiang, C., Karhunen, V., Lu, Y., Lu, Q., Mallard, T. T., Mishra, P. P., Nolte, I. M., Palviainen, T., Peterson, R. E., Sallis, H. M., Shabalin, A. A., Tate, A. E., ... Middeldorp, C. M. (2022). Genome-wide Association Meta-analysis of Childhood and Adolescent Internalizing Symptoms. *Journal of the American Academy of Child & Adolescent Psychiatry*, 61(7), 934–945. <https://doi.org/10.1016/j.jaac.2021.11.035>
- Janda, J. M., & Abbott, S. L. (2007). 16S rRNA Gene Sequencing for Bacterial Identification in the Diagnostic Laboratory: Pluses, Perils, and Pitfalls. *Journal of Clinical Microbiology*, 45(9), 2761–2764. <https://doi.org/10.1128/jcm.01228-07>
- Kelsey, C. M., Prescott, S., McCulloch, J. A., Trinchieri, G., Valladares, T. L., Dreisbach, C., Alhusen, J., & Grossmann, T. (2021). Gut microbiota composition is associated with newborn functional brain connectivity and behavioral temperament. *Brain, Behavior, and Immunity*, 91(August 2020), 472–486. <https://doi.org/10.1016/j.bbi.2020.11.003>
- Khalatbari-Soltani, S., Maccora, J., Blyth, F. M., Joannès, C., & Kelly-Irving, M. (2022). Measuring education in the context of health inequalities. *International Journal of Epidemiology*, 51(3), 701–708. <https://doi.org/10.1093/ije/dyac058>
- Kline, R. B. (2016). *Principles and practice of structural equation modeling, 4th ed* (pp. xvii, 534). The Guilford Press.
- Le Cao, K.-A. (2023, March). *Continuous response variable Y in DIABLO?* [Online post]. MixOmics User Forum. <https://mixomics-users.discourse.group/t/continuous-response-variable-y-in-diablo/58/5>

- Lê Cao, K.-A., Martin, P. G., Robert-Granié, C., & Besse, P. (2009). Sparse canonical methods for biological data integration: Application to a cross-platform study. *BMC Bioinformatics*, *10*(1), 34. <https://doi.org/10.1186/1471-2105-10-34>
- Le Cao, K.-A., & Welham, Z. (2022). *Multivariate data integration using R: Methods and Applications with the MixOmics package*. CRC Press.
- Lindberg, M. H., Chen, G., Olsen, J. A., & Abelsen, B. (2022). Combining education and income into a socioeconomic position score for use in studies of health inequalities. *BMC Public Health*, *22*(1), 969. <https://doi.org/10.1186/s12889-022-13366-8>
- MacCallum, R. C., & Austin, J. T. (2000). Applications of structural equation modeling in psychological research. *Annual Review of Psychology*, *51*, 201–226. <https://doi.org/10.1146/annurev.psych.51.1.201>
- Meinshausen, N., & Bühlmann, P. (2010). Stability selection. *Journal of the Royal Statistical Society: Series B (Statistical Methodology)*, *72*(4), 417–473. <https://doi.org/10.1111/j.1467-9868.2010.00740.x>
- Mihalik, A., Chapman, J., Adams, R. A., Winter, N. R., Ferreira, F. S., Shawe-Taylor, J., & Mourão-Miranda, J. (2022). Canonical Correlation Analysis and Partial Least Squares for Identifying Brain–Behavior Associations: A Tutorial and a Comparative Study. *Biological Psychiatry: Cognitive Neuroscience and Neuroimaging*, *7*(11), 1055–1067. <https://doi.org/10.1016/j.bpsc.2022.07.012>
- Solmi, M., Radua, J., Olivola, M., Croce, E., Soardo, L., Salazar de Pablo, G., Il Shin, J., Kirkbride, J. B., Jones, P., Kim, J. H., Kim, J. Y., Carvalho, A. F., Seeman, M. V., Correll, C. U., & Fusar-Poli, P. (2022). Age at onset of mental disorders worldwide: Large-scale meta-analysis of 192 epidemiological studies. *Molecular Psychiatry*, *27*(1),

281–295. <https://doi.org/10.1038/s41380-021-01161-7>

Song, X., Niu, L., Admon, R., Long, J., Li, Q., Peng, L., Lee, T. M. C., & Zhang, R. (2024).

Aberrant positive affect dynamics in individuals with subthreshold depression: Evidence from laboratory and real-world assessments. *International Journal of Clinical and Health Psychology*, *24*(1). <https://doi.org/10.1016/j.ijchp.2023.100427>

Tang, S., Lu, L., Zhang, L., Hu, X., Bu, X., Li, H., Hu, X., Gao, Y., Zeng, Z., Gong, Q., &

Huang, X. (2018). Abnormal amygdala resting-state functional connectivity in adults and adolescents with major depressive disorder: A comparative meta-analysis. *eBioMedicine*, *36*, 436–445. <https://doi.org/10.1016/j.ebiom.2018.09.010>

van de Wouw, M., Wang, Y., Workentine, M. L., Vaghef-Mehrabani, E., Barth, D., Mercer, E.

M., Dewey, D., Arrieta, M.-C., Reimer, R. A., Tomfohr-Madsen, L., & Giesbrecht, G. F.

(2024). Cluster-specific associations between the gut microbiota and behavioral outcomes in preschool-aged children. *Microbiome*, *12*(1), 60. <https://doi.org/10.1186/s40168-024-01773-5>

van de Wouw, M., Wang, Y., Workentine, M. L., Vaghef-Mehrabani, E., Dewey, D., Reimer, R.

A., Tomfohr-Madsen, L., & Giesbrecht, G. F. (2022). Associations Between the Gut

Microbiota and Internalizing Behaviors in Preschool Children. *Psychosomatic Medicine*,

84(2), 159. <https://doi.org/10.1097/PSY.0000000000001026>

Vergunst, F., Comisso, M., Geoffroy, M.-C., Temcheff, C., Poirier, M., Park, J., Vitaro, F.,

Tremblay, R., Côté, S., & Orri, M. (2023). Association of Childhood Externalizing,

Internalizing, and Comorbid Symptoms With Long-term Economic and Social Outcomes.

JAMA Network Open, *6*(1), e2249568.

<https://doi.org/10.1001/jamanetworkopen.2022.49568>

Webb, C. A., Israel, E. S., Belleau, E., Appleman, L., Forbes, E. E., & Pizzagalli, D. A. (2021).

Mind-Wandering in Adolescents Predicts Worse Affect and Is Linked to Aberrant Default Mode Network–Salience Network Connectivity. *Journal of the American Academy of Child & Adolescent Psychiatry*, *60*(3), 377–387.

<https://doi.org/10.1016/j.jaac.2020.03.010>

Zhang, R., Peng, X., Song, X., Long, J., Wang, C., Zhang, C., Huang, R., & Lee, T. M. C.

(2022). The prevalence and risk of developing major depression among individuals with subthreshold depression in the general population. *Psychological Medicine*, *53*(8), 3611–3620. <https://doi.org/10.1017/S0033291722000241>

Reviewer #4:

I appreciate that the authors have addressed most of my previous suggestions. However, I was left wondering why some of the supplemental analyses and results are reported only in the Methods section and the Supplementary Information (SI). I recommend moving the description of the predicted functional pathways to the Results section to ensure it does not escape readers' attention. The authors have explained this aspect in more detail in their response letter.

We thank the reviewer for this recommendation. We have added a description of the predicted functional pathways to the manuscript Results section.

Starting line 403: *“Predicted Microbiota Functional Profiles That Maximally Covaried with Brain Signatures*

To better understand relations between the predicted functional potential of the microbiota and identified brain signatures, we estimated the abundance of functional pathways from the 16S sequencing data using PiCRUST 2, and then performed sPLS analyses with that predicted functional data as input. We found that one predicted functional profile covaried most strongly with the SOFA, MTL, SAL, PMN Network Connectivity Brain Signature, and remained significantly associated with its corresponding brain signature when controlling for covariates. The pathway that had high stability from this component, which also had the strongest loading, was PWY-6270, isoprene biosynthesis I (negative loading). We also found two predicted functional profiles covaried most strongly with the SOFA Between Network Connectivity Brain Signature, one of which remained significantly associated with its corresponding brain signature when controlling for covariates. Pathways from these profiles with the highest loadings, and high stability, included the pentose phosphate pathway and superpathway of glucose and xylose degradation (negative loadings), and cob(II)yrinate a,c-diamide biosynthesis I pathway, NAD salvage pathways V and I, and UPD-N-acetyl-D-glucosamine biosynthesis I pathway (positive loadings). There were no significant total, direct, or indirect associations (via the corresponding brain signature) between the functional profiles and caregiver-reported internalizing symptoms at age 7.5 years. Full methods and results are presented in the Supplementary Methods and Notes, respectively. ”

Regrettably, there is little that can be done about the low sample size, and having specified hypothesis as presented here definitely is a merit. However, the low sample size may lead to difficulty in testing whether the current interpretations would hold if alternative, complementary data analysis methods were applied.

We agree with the reviewer that the small sample size, while similar to other papers on this topic, is an important limitation. We definitely recommend that future research attempts to replicate our findings with alternative, complementary methods in larger samples, and have added an additional clause to the limitations section to further emphasize this important point.

Starting line 583: *“Our findings should be replicated in larger samples with separate training and testing sets, and using complementary data analysis methods, to evaluate generalizability.”*